# CLEM-Reg: an automated point cloud-based registration algorithm for volume correlative light and electron microscopy

**Daniel Krentzel** [1,2] ✉, **Matouš Elphick** [2,3,4,5], **Marie-Charlotte Domart**[2], **Christopher J. Peddie** [2], **Romain F. Laine**[6,9], **Cameron Shand** [7], **Ricardo Henriques** [6,8], **Lucy M. Collinson** [2] & **Martin L. Jones** [2] ✉

Volume correlative light and electron microscopy (vCLEM) is a powerful imaging technique that enables the visualization of fluorescently labeled proteins within their ultrastructural context. Currently, vCLEM alignment relies on time-consuming and subjective manual methods. This paper presents CLEM-Reg, an algorithm that automates the three-dimensional alignment of vCLEM datasets by leveraging probabilistic point cloud registration techniques. Point clouds are derived from segmentations of common structures in each modality, created by state-of-the-art open-source methods. CLEM-Reg drastically reduces the registration time of vCLEM datasets to a few minutes and achieves correlation of fluorescent signal to submicron target structures in electron microscopy on three newly acquired vCLEM benchmark datasets. CLEM-Reg was then used to automatically obtain vCLEM overlays to unambiguously identify TGN46-positive transport carriers involved in protein trafficking between the trans-Golgi network and plasma membrane. Datasets are available on EMPIAR and BioStudies, and a napari plugin is provided to aid end-user adoption.

Correlative light and electron microscopy (CLEM) is a powerful imaging technique that seeks to capitalize on the advantages of light microscopy (LM) and electron microscopy (EM) while circumventing the drawbacks of each. This has made CLEM the imaging technique of choice to target rare and dynamic biological events that require structural analysis at high resolution[1,2]. Fluorescence microscopy (FM) is an LM imaging modality that generates contrast by tagging macromolecules in living cells and tissues with fluorescent proteins, enabling dynamic observation of their biological interactions; however, due to the diffraction limit of light, traditional FM cannot achieve a resolution better than around 200 nm, hindering fine structural details from being resolved[3]. While super-resolution techniques can surpass this diffraction limit, such methods require specialized instruments, specific sample preparation and imaging protocols, imposing additional constraints on the type of biological events that can be imaged[4]. Moreover, FM generally tags specific macromolecules, providing excellent molecular specificity; however, unlabeled structures cannot be observed. EM addresses these limitations, achieving orders of magnitude higher resolution while revealing the underlying biological context[5] in exquisite detail, but at the cost of a smaller field of view (FOV) and the lack of molecular specificity. By harnessing the complementary information derived from the correlation of LM and EM, CLEM has led to a variety of biological discoveries, such as establishing the structure of tunneling nanotubes in neurons[6], observing

[1]Imaging and Modeling Unit, Institut Pasteur, Université Paris Cité, Paris, France. [2]Electron Microscopy Science Technology Platform, The Francis Crick Institute, London, UK. [3]Cancer Dynamics Laboratory, The Francis Crick Institute, London, UK. [4]Department of Bioengineering, Imperial College London, London, UK. [5]The Institute of Cancer Research, London, UK. [6]UCL-Laboratory for Molecular Cell Biology, University College London, London, UK. [7]Software Engineering & AI Science Technology Platform, The Francis Crick Institute, London, UK. [8]Instituto de Tecnologia Química e Biológica António Xavier, Universidade Nova de Lisboa, Oeiras, Portugal. [9]Present address: Abbelight, Cachan, France. ✉e-mail: daniel.krentzel@pasteur.fr; martin.jones@crick.ac.uk

blood vessel fusion events in zebrafish[2] and localizing tuberculosis bacteria in primary human cells[1].

Typically, CLEM data are obtained by sequentially imaging a sample in FM and then EM. First, relevant structures are tagged with organic dyes or fluorescent proteins, and a volumetric image stack is acquired using FM. The sample is fixed with crosslinkers, either before or after the FM imaging, to conserve structural features. It is then stained with heavy metal salts to introduce contrast, dehydrated, embedded in resin and trimmed to the region of interest (ROI)[7]. In volume EM (vEM), layers of the embedded sample are physically removed and either the face of the block or the sections themselves are imaged in EM to obtain an image volume[5]. This results in two corresponding image stacks, one from FM and one from EM, each containing complementary data from the same physical region of the sample, but typically imaged in different orientations. In addition to this orientation mismatch, the sample preparation and imaging can introduce both linear and nonlinear deformations between the FM and EM image volumes. To correlate the FM signal to the ultrastructure in EM, image volumes need to be registered. Due to the stark differences in resolution, contrast and FOV between FM and EM, this is a challenging task that cannot be approached with intensity-based methods that are routinely used for aligning data from visually similar modalities, for example magnetic resonance imaging (MRI) and computed tomography (CT).

There are two general approaches to solving this problem. The first approach is to process one or both images such that they share a similar visual appearance, for example, by directly converting across modalities[8-10] or by constructing a shared modality-agnostic representation[11,12]. Once the processed image stacks are sufficiently similar in visual appearance, traditional intensity-based registration techniques, such as those employed in medical imaging[13], can be used to automatically align the two datasets. The second approach uses a landmark-based method, such as those implemented in software tools like BigWarp[14] and eC-CLEM[15]. These tools rely on manually identifying precise spatial regions visible in both modalities, for example, small subcellular structures or prominent morphological features. By manually placing a landmark at an identical physical position in each modality, spatial correspondences can be established. From these landmarks, the transformation between image volumes can be computed[1], bringing them into alignment via an iterative optimization process[16,17]. Methods to automate alignment via detection of cell centroids in LM and EM[18] or semi-automated feature detection (for example AutoFinder in eC-CLEM[15]) have been developed; however, such methods are often restricted to two dimensions or limited to relatively coarse alignment, requiring subsequent manual refinement. Moreover, deep-learning-based methods that convert across modalities[8] or construct a shared modality-agnostic representation[12] require large amounts of aligned ground truth data. Due to the low throughput and required expertise of manual volume CLEM (vCLEM) alignment, generating such ground truth data is challenging.

An important advantage of landmark-based approaches is the inherently sparse representation, which substantially reduces memory and computational requirements compared to intensity-based registration techniques that must generally hold both image volumes in working memory; however, this manual landmark selection step is laborious and time-consuming, severely impacting throughput and potentially introducing bias, as the target structure may be directly used for registration. To avoid such biases, landmarks used for registration should be different from the target structures being studied wherever possible, taking care to ensure color-correction between the channels in FM to avoid spectrally induced shifts in focal depth. Due to these limitations, robust and objective automation of landmark detection is highly desirable.

Here, CLEM-Reg is introduced, an automated vCLEM registration algorithm that relies on extracting landmarks from common structures in each modality. To achieve this aim, a workflow to segment mitochondria was developed. Mitochondria were chosen specifically because they are abundant and typically well distributed across cells, and easily imaged in both FM and EM, enabling robust matching across modalities. Various segmentation approaches are routinely used in microscopy, ranging from classical image processing techniques[9,19,20] to machine learning[21-23]. Depending on the complexity and density of structures in the image at hand, different algorithms are appropriate. For instance, mitochondria segmentations in FM images can be obtained by filtering, thresholding and further downstream image processing[19], while automatically segmenting EM data typically requires deep learning. Common deep-learning architectures such as 'U-Net'[24] can be trained to very good effect, but the burden of obtaining sufficient ground truth data presents a huge challenge, often requiring substantial amounts of expert effort or crowdsourcing of manual annotations[23]. Recently, however, pretrained 'generalist' models such as MitoNet[25], based upon a 'Panoptic-DeepLab' architecture[26], are able to provide out-of-the-box performance levels for mitochondrial segmentation in EM that are sufficient for many tasks, with the option to fine-tune where necessary.

After segmentation, points are equidistantly sampled from the surface of the mitochondria segmentations in both the FM and EM volumes, resulting in a 'point cloud' for each modality. Point clouds are an attractive modality-agnostic representation due to their inherent sparsity and the availability of a range of performant registration algorithms[16,27-29]; however, unlike manually generated pairs of points, there is no guarantee of a one-to-one precise spatial correspondence between points in the different modalities. The coherent point drift (CPD)[27] algorithm overcomes this limitation by casting the alignment task as a probability density estimation problem, thereby removing the constraint of strict point correspondences.

Assessing registration performance in vCLEM overlays is challenging. Unlike in the medical imaging field where multimodal registration of MRI and CT scans is achieved by minimizing mutual information (MI)[30], no equivalent metrics exist to compare FM and EM volumes. This lack of metrics is an important hurdle in automating vCLEM alignment and evaluating registration performance. Currently, vCLEM overlay quality is assessed by experts via visual inspection, oftentimes by the same person that generated the alignment.

The aim of vCLEM experiments is to functionally label ultrastructure in EM with fluorescent signal. These structures can range from microns to a few nanometers in size. To test the limits of CLEM-Reg, registration performance was assessed on some of the smallest known organelles, namely lysosomes, which have a size of 0.3–1 μm (refs. [31,32]). To quantify registration performance, two new metrics based on the correlation of fluorescent signal (LysoTracker) to submicron target structures (lysosomes) in EM are introduced here. Specifically, the volume of lysosomes overlaid by fluorescence was computed and centroid distances between fluorescent signal and lysosomes calculated. Manual registration by an expert was used as a baseline for assessing the performance of CLEM-Reg.

Performance quantification was conducted on two newly acquired vCLEM benchmark datasets using two different EM modalities: focused ion beam scanning electron microscope (FIB-SEM) and serial block-face scanning electron microscope (SBF-SEM). A third vCLEM dataset was acquired to investigate a rare and dynamic cellular process involving submicron organelles. TGN46 (TGN38 in rodents) has previously been observed in transport carriers involved in protein trafficking between the trans-Golgi network (TGN) and plasma membrane[33-36]. These transport carriers are rare, as they account for only 1–5% of the total TGN46 signal[36]. Here, CLEM-Reg is used to automatically register the GFP-TGN46 signal to EM ultrastructure in three dimensions, facilitating accurate identification of TGN46 transport carriers that would have been missed by visual inspection of the EM volume alone.

## Results

### Benchmark dataset acquisition

To assess the performance of CLEM-Reg against an expert, three benchmark vCLEM datasets (EMPIAR-10819, EMPIAR-11537 and EMPIAR-11666) of human cervical cancer epithelial (HeLa) cells were acquired. Mitochondria (MitoTracker Deep Red), nucleus (Hoechst 33342), Golgi apparatus protein TGN46 (GFP-TGN46) and lysosomes (LysoTracker red) in EMPIAR-10819 and EMPIAR-11666 and plasma membrane (WGA) in EMPIAR-11537 were tagged to enable unbiased registration performance assessment on target structures. After imaging the samples with a Zeiss Airyscan LSM900 microscope, two corresponding EM volumes (EMPIAR-10819 and EMPIAR-11537) with an isotropic voxel size of 5 nm were acquired in a FIB-SEM. The EM volume in EMPIAR-11666 was acquired in an SBF-SEM (7 nm in $xy$ and 50 nm in $z$). The acquired images were prealigned to the nearest orthogonal rotation following a routine image processing workflow in Fiji (Methods and Fig. 1a).

### The CLEM-Reg pipeline

CLEM-Reg automatically aligns vCLEM data by segmenting mitochondria in FM and EM, generating point clouds and registering them with CPD[27,28], a state-of-the-art point cloud registration technique. The FM volume is then warped onto the EM volume using the found transformation (Fig. 1b). To aid adoption, CLEM-Reg is deployed as a plugin ('napari-clemreg') for the napari image viewer[37], giving users the option for a single-click end-to-end operation, or to fine-tune or even entirely replace individual workflow steps (for example, importing segmentations; Fig. 1c).

### Segmenting internal landmarks

A promising approach to automating vCLEM registration is to automatically identify internal landmarks, speeding up the process and minimizing inadvertent subjective bias. CLEM-Reg relies on segmenting these common internal landmarks in both imaging modalities. Here, mitochondria were used as landmarks.

To obtain segmentations in FM, an algorithm based on combining a three-dimensional (3D) Laplacian of Gaussian (LoG) filter with dynamic thresholding to account for signal degradation at deeper imaging levels was developed. The algorithm requires two parameters to be adjusted: kernel size and relative segmentation threshold. After obtaining an initial segmentation mask, spurious segmentations are removed with a size-based filter (Fig. 2a).

Mitochondria segmentations in EM are obtained with a pretrained MitoNet[25] deep-learning model, which was found to perform well on FIB-SEM data out-of-the-box (Fig. 2b) and required slight preprocessing for SBF-SEM data (Methods). CLEM-Reg's robustness to missing mitochondria segmentations in the EM volume was estimated by randomly removing segmentations (Extended Data Fig. 1a). The registration accuracy of CLEM-Reg was constant up to a loss of around 40% of segmented mitochondria in EM. The impact of segmentation errors in different areas of the EM volume was also assessed (Extended Data Fig. 1b). Registration performance was most impacted by the loss of peripheral segmentations. While mitochondria were used as off-target landmarks here, note that CLEM-Reg is not restricted to using mitochondria segmentations and can be used with previously obtained EM segmentation masks of other structures or organelles (for example, nuclear envelope).

### Generating modality-agnostic point clouds and registration

The alignment between the FM and EM segmentations can be inferred by sampling 3D point clouds from the previously obtained segmentation masks. This reduces the computational load for large datasets and allows for mistakes in the segmentation to be ignored by using a probabilistic registration algorithm, such as CPD. The extraction of pixel coordinates from the exterior of the segmentation masks results

in a 3D point cloud. The number of points in both point clouds depends on two parameters: binning and downsampling factor (Fig. 3a). Increasing any of these two parameters speeds up the registration, potentially by orders of magnitude. For instance, reducing the point sampling frequency from 1/16 to 1/256 with a fixed voxel size of 10 × 10 × 10 pixels (points within each voxel are averaged to generate one point) leads to a 19-fold decrease (from 33.7 min to 1.8 min) in registration time with no change in registration performance on EMPIAR-10819 (Extended Data Fig. 2). Indeed, the time required to register point clouds follows a power law (Extended Data Fig. 3) with an exponent ranging between 1.47 and 1.69 which implies that doubling the number of sampled points increases the time required for registration by a factor of $2^{1.47}$ to $2^{1.69}$.

After sampling, the point clouds are registered using either rigid CPD, affine CPD or nonlinear Bayesian CPD (BCPD) (Fig. 3b)[27]. Note that these probabilistic methods are necessary[29], as the sampled points are not paired across modalities. The choice of registration algorithm depends on the expected deformations between the FM and EM volumes, as well as computational constraints. In general, rigid CPD is faster and computationally less expensive than nonlinear BCPD.

### Warping FM volume to obtain vCLEM overlay

Once point clouds are registered, the found transformation is used to warp the FM volume onto the EM volume. This step is fast for rigid transformations but orders of magnitude slower for nonlinear warping. CLEM-Reg implements 3D nonlinear thin-plate spline warping[38] that uses the initially sampled and registered FM point clouds as control points. The runtime of the thin-plate spline warping depends on the interpolation order and size of the approximate grid. As thin-plate spline warping is an expensive algorithm, CLEM-Reg also implements the option to sequentially warp subvolumes. While this extends the runtime of the warping step, it reduces the required random-access memory (RAM). Notably, overlays obtained with rigid alignment (Fig. 4a–c) outperformed nonlinear alignment on the three benchmark datasets (Extended Data Fig. 4).

### Assessing CLEM-Reg performance against experts

Due to the lack of existing metrics for vCLEM alignment, two metrics were introduced to holistically assess registration performance: fluorescent signal overlap to EM structures and centroid distance between fluorescent signal and target structures. Additional quantification was conducted on manually placed landmarks in the LM and EM volumes.

For the correlation of fluorescence to FIB-SEM data, five target structures (lysosomes) were manually segmented in EM. The selected lysosomes varied in size (0.029–0.522 μm³) and were distributed across the cell volume (Supplementary Fig. 1a,b).

To obtain CLEM-Reg overlays of the LysoTracker channel, off-target landmarks (mitochondria) were used for registration. This reduces bias that results from aligning fluorescence directly to presumed target structures. Registration with CLEM-Reg took 5.52 min (from starting registration to obtaining warped overlays for all channels, including mitochondria segmentation) on a portable machine (Methods). Manual registration was performed with BigWarp by an expert with access to all fluorescent channels requiring approximately 2 h (Fig. 5a).

The volume of segmented lysosomes in EM overlaid by LysoTracker signal was computed by first segmenting the LysoTracker channel with Otsu thresholding[39] and then computing the intersected volume between both segmentations. It was found that all lysosomes segmented in EM were overlaid with the LysoTracker signal regardless of size (Fig. 5b). Notably, even the smallest lysosome with a volume of 0.029 μm³ was labeled with fluorescence, showing correct correlation well below the diffraction limit.

Next, centroid distances between segmented LysoTracker signal and lysosome segmentations were computed. Lysosomes were on average 2.62 times larger than average centroid distances obtained with CLEM-Reg, indicating unambiguous labeling (Fig. 5c). Average

**a** vCLEM data generation

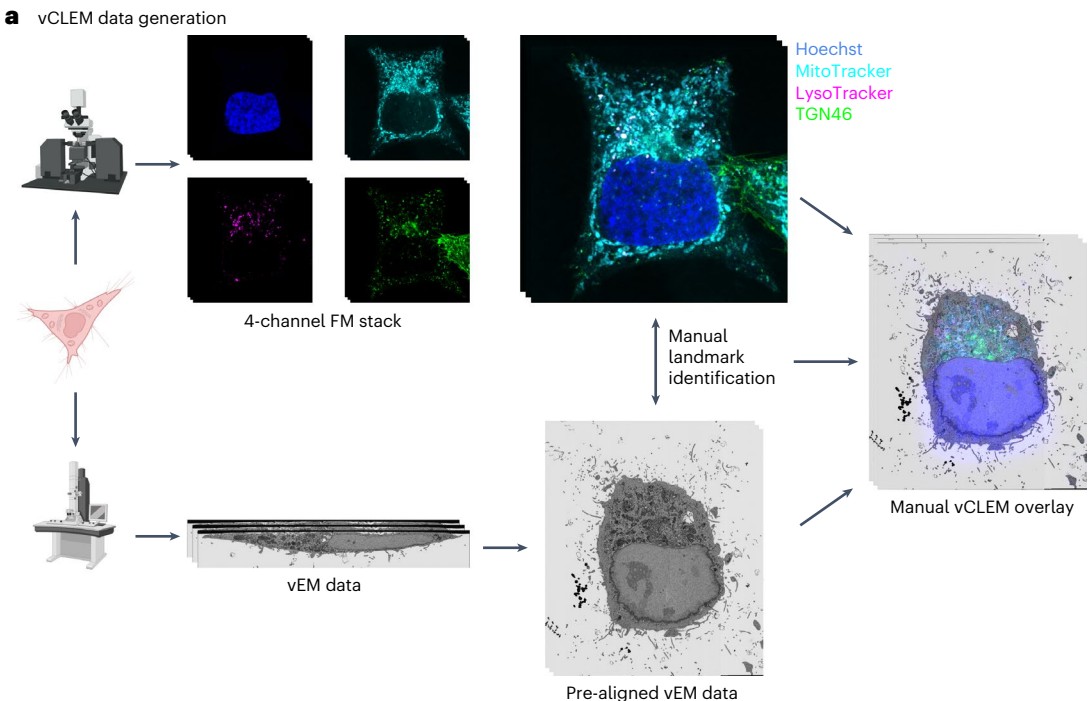

**b** Automated vCLEM registration with CLEM-Reg

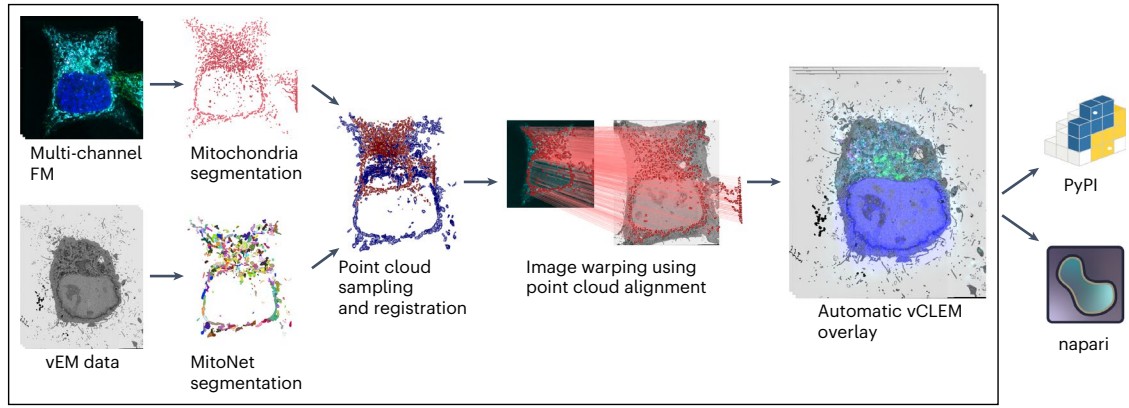

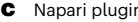

**c** Napari plugin

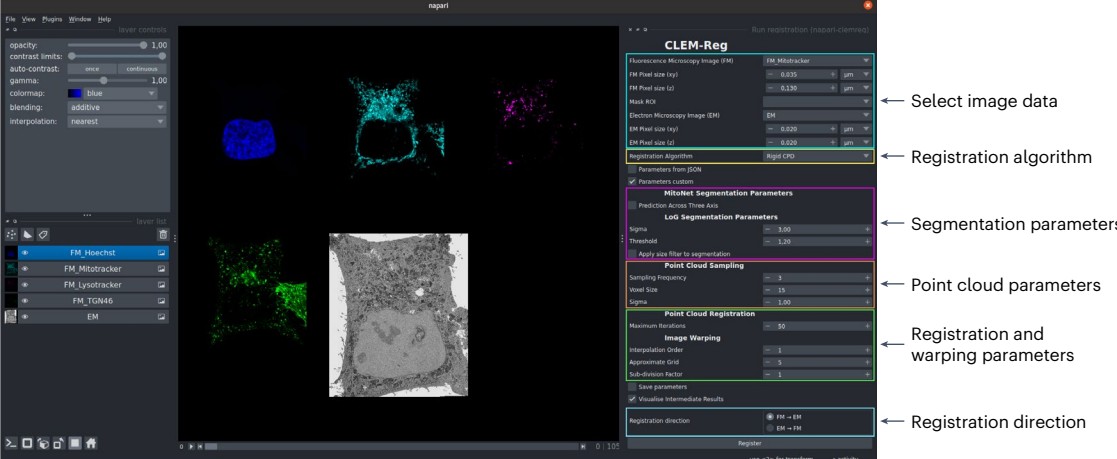

**Fig. 1 | Volume CLEM data generation and CLEM-Reg algorithm. a,** Obtaining a vCLEM dataset consists of acquiring a FM and vEM image of the same sample. The two image stacks are traditionally manually aligned by identifying landmarks in both modalities and computing a transform to warp the FM stack onto the vEM data. **b,** CLEM-Reg fully automates the registration step for vCLEM datasets by first segmenting mitochondria in both image modalities, sampling point clouds from these segmentations and registering them. Once registered, the point cloud alignment is used to warp the FM stack onto the vEM data. All data visualizations were generated with napari. **c,** The napari-clemreg plugin automatically registers vCLEM datasets with a single button click. Panel **a** created with Biorender.com.

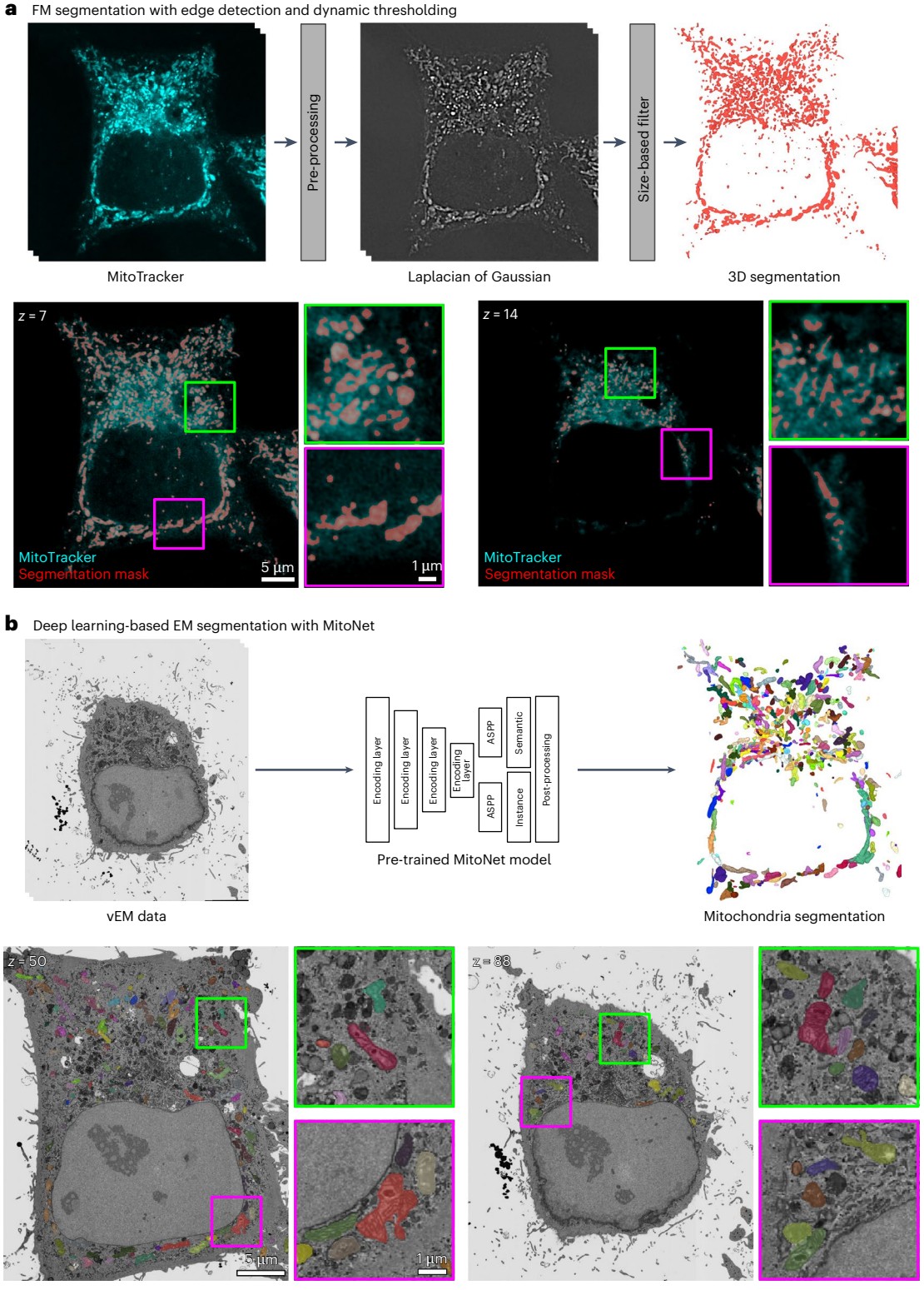

**Fig. 2 | Mitochondria segmentation in FM and EM. a**, Mitochondria in the MitoTracker channel are segmented by applying a 3D LoG filter to extract edges and dynamically thresholded to account for decreasing pixel intensity values as the imaging depth increases. To remove spurious mitochondria segmentations, a size-based filter is used. **b**, Mitochondria in the vEM data are segmented with a pretrained MitoNet[25] model. The MitoNet architecture is composed of four encoding layers, two atrous spatial pyramid pooling layers followed by semantic and instance segmentation outputs, which are post-processed to yield a final panoptic segmentation mask shown on the right-hand side. Visualizations were generated with napari.

centroid distances obtained with CLEM-Reg were within 100.98 nm of centroid distances obtained from manual registration and thus below the theoretical resolution of the FM which, for the imaging system used, was 120 nm in $xy$ and 350 nm in $z$ (Fig. 5d). 3D visualization overlays of

segmentations and centroids of each lysosome are shown in Fig. 5e. Further quantifications demonstrating equivalent performance on conventional resolution confocal microscopy data (Supplementary Fig. 2a,b) are shown in Extended Data Fig. 5a–d.

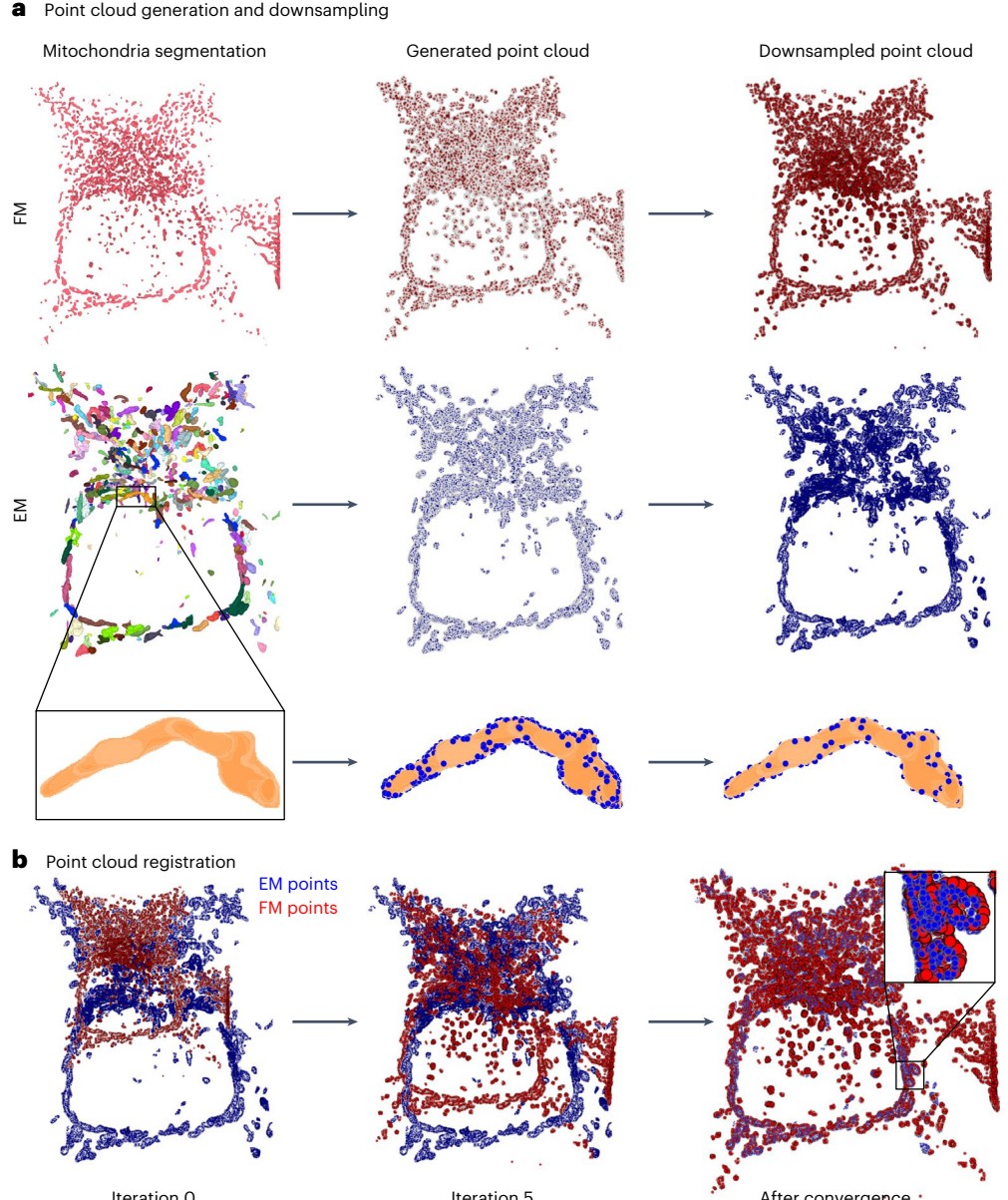

**a** Point cloud generation and downsampling

Mitochondria segmentation | Generated point cloud | Downsampled point cloud

**b** Point cloud registration

EM points
FM points

Iteration 0 | Iteration 5 | After convergence

**Fig. 3 | Point cloud generation and registration. a**, Point clouds are sampled on the surface of the 3D mitochondria segmentations in both FM and EM. To reduce the computational load and speed up the alignment time (Extended Data Figs. 2 and 3), both point clouds are downsampled. **b**, The point clouds are registered using rigid CPD[28] until convergence (50 iterations). Visualizations were generated with napari.

To assess the generalizability of CLEM-Reg to other EM modalities, performance was additionally assessed on a dataset acquired in SBF-SEM. CLEM-Reg overlays were obtained by registering mitochondria on a portable machine (Methods) requiring 2.44 min including mitochondria segmentation and image warping, while manual registration with BigWarp took around 2 h (Fig. 5f). A total of four lysosomes distributed across the cell were manually segmented for performance quantification (Supplementary Fig. 1c–f).

CLEM-Reg overlaid LysoTracker signal to all four lysosomes in the SBF-SEM volume regardless of size (Fig. 5g). Lysosomes were on average 6.95 times larger than centroid distances obtained with CLEM-Reg, indicating unambiguous correlation between fluorescent signal and lysosomes (Fig. 5h). On average, centroid distances in the overlay obtained with CLEM-Reg were within 54.88 nm of centroid distances obtained from manual registration. Notably, CLEM-Reg achieved smaller centroid distances compared to the manual registration on two lysosomes (Fig. 5i). 3D visualization overlays of segmentations and

centroids of each lysosome are shown in Fig. 5j. In addition, equivalent alignment performance of CLEM-Reg on conventional resolution confocal microscopy data (Supplementary Fig. 2e,f) was verified in Extended Data Fig. 5e–h.

To evaluate whether alignment performance depended on proximity of target structures to segmented mitochondria, centroid distances between fluorescent signal and target structures were correlated with distances of target structures (Extended Data Fig. 6 shows quantification of endosome overlay on EMPIAR-11537) to mitochondria using Spearman's correlation (Extended Data Fig. 7). Only a slight correlation of $\rho = 0.5$ in one dataset (EMPIAR-10819) and no correlation in two datasets (EMPIAR-11537 and EMPIAR-11666) could be observed, indicating that proximity to mitochondria did not lead to improved registration accuracy.

Next, performance of CLEM-Reg against manual registration was quantified on manually placed landmarks in the LM and EM volumes using BigWarp (EMPIAR-10819, number of point pairs $n = 145$;

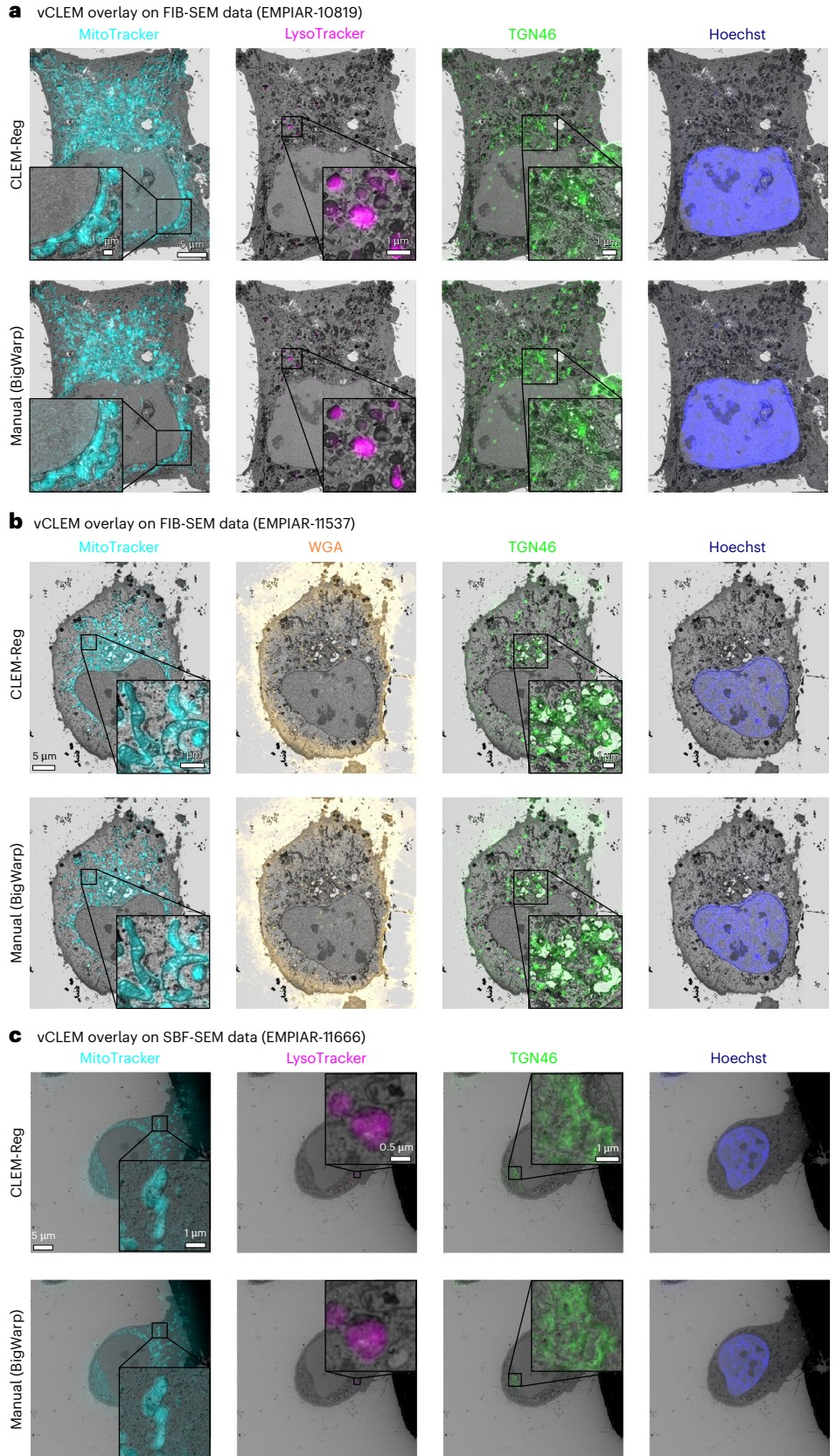

**Fig. 4 | Comparing overlays obtained manually and with CLEM-Reg.** CLEM-Reg overlays were obtained with rigid registration using the napari-clemreg plugin, while the manual overlays were obtained with affine registration using the BigWarp plugin in Fiji. **a**, vCLEM overlays for EMPIAR-10819 dataset showing mitochondria (MitoTracker), lysosomes (LysoTracker), Golgi apparatus (TGN46) and nucleus (Hoechst) with EM data acquired on FIB-SEM microscope. **b**, vCLEM overlays for EMPIAR-11537 dataset showing MitoTracker, WGA, GFP-TGN46 and Hoechst with EM data acquired on FIB-SEM microscope. **c**, vCLEM overlays for EMPIAR-11666 dataset showing MitoTracker, LysoTracker, GFP-TGN46 and Hoechst staining with EM data acquired on SBF-SEM microscope. Overlays were generated with napari.

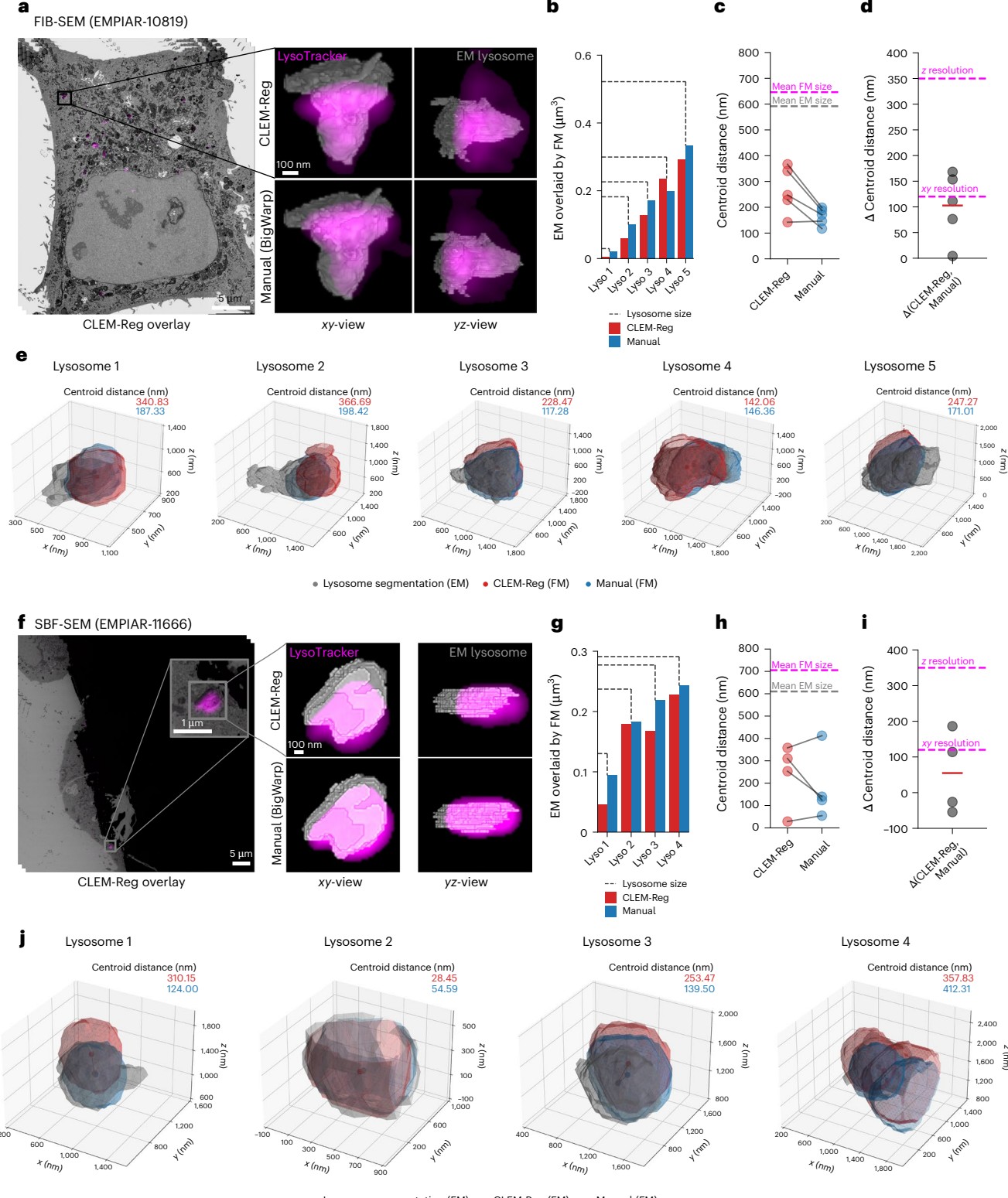

**Fig. 5 | Comparing alignment results between CLEM-Reg and experts.**
**a,f**, LysoTracker channels were overlaid to FIB-SEM (EMPIAR-10819) and SBF-SEM (EMPIAR-11666) data using mitochondria as off-target landmarks with CLEM-Reg. To quantify registration performance, five lysosomes were manually segmented throughout each EM volume. Corresponding segmentations in FM were obtained by segmenting the LysoTracker channel with Otsu's method[39]. **b,g**, Volume of lysosomes in EM overlaid by FM signal was computed by intersecting EM and FM segmentations. **c,h**, Centroid distances between EM segmentations and segmented LysoTracker signal in FM were computed with Euclidean distance. Mean size of FM ($n$ = 5 lysosomes in **c** and $n$ = 4 lysosomes

in **h**) and EM ($n$ = 5 lysosomes in **c** and $n$ = 4 lysosomes in **h**) segmentations are shown in magenta and gray, respectively. **d,i**, The difference between LysoTracker signal overlaid manually and with CLEM-Reg was computed from previously found centroid distances. Mean difference in centroid distances ($n$ = 5 lysosomes in **d** and $n$ = 4 lysosomes in **i**) is shown with a red horizontal line. The theoretical $xy$ and $z$ resolution of the fluorescence microscope used is shown in magenta. **e,j**, 3D visualizations of lysosome overlays were generated by obtaining meshes from segmentations of EM shown in gray, LysoTracker signal registered using BigWarp (Manual) shown in blue and CLEM-Reg shown in red. Visualizations were generated with napari and matplotlib.

EMPIAR-11537, $n = 42$; and EMPIAR-11666, $n = 55$). Landmarks placed in the LM volume were transformed with rigid and affine warping matrices obtained from CLEM-Reg and BigWarp, respectively. Then, Euclidean distances between transformed landmarks placed in the LM volume and corresponding landmarks placed in the EM volume were computed (Extended Data Fig. 8a–c and Extended Data Fig. 9 for qualitative error maps between landmarks placed in EM and LM landmarks transformed with CLEM-Reg). Distances between EM landmarks and LM landmarks transformed by warping matrices found with CLEM-Reg and BigWarp were not significantly different (Student's $t$-test) in one out of three datasets (EMPIAR-10819, $P > 0.05$; EMPIAR-11537, $P < 0.01$; and EMPIAR-11666, $P < 0.0001$). Note, however, that warping matrices obtained from BigWarp were computed to explicitly minimize distances between manually placed landmarks in LM and EM. Thus, Euclidean distances computed between EM landmarks and LM landmarks transformed by warping matrices obtained from BigWarp correspond to the residual registration error which arises due to landmarks being placed by an expert relying on visual inspection only. To explore the robustness of manual landmark placement, landmarks placed in the LM and EM volumes were randomly offset by up to 10 pixels (Gaussian random noise with $\mu = 0$ and $\sigma = n_{pixels}$) and distances between LM landmarks transformed by CLEM-Reg and BigWarp computed as described above (Extended Data Fig. 8d). Notably, randomly offsetting landmarks by up to 3 (EMPIAR-11537) and 5 (EMPIAR-11666) pixels was sufficient to obtain nonsignificant ($P > 0.05$ with Student's $t$-test) differences in alignment between CLEM-Reg and manual registration (Extended Data Fig. 8e,f).

Overall, these results indicate that CLEM-Reg successfully automates vCLEM registration of both FIB-SEM and SBF-SEM data, correlating submicron structures with near expert-level accuracy while considerably reducing registration time. Moreover, fluorescent overlay of target structures does not correlate with distance to segmented off-target structures (mitochondria) and registration with CLEM-Reg is equivalent to BigWarp assuming manual landmarks are placed with an error not exceeding five pixels. Crucially, registration with CLEM-Reg was performed using an off-target channel (MitoTracker) and assessed on an unseen target channel (LysoTracker) reducing potential biases arising from directly aligning target structures.

### CLEM-Reg plugin for napari

Napari is an open-source multidimensional image viewer for Python. It allows third-parties to develop plugins with additional custom functionality[37]. The plugin allows users to automatically register vCLEM datasets with a single click of a button. It also includes the option to execute and display intermediate steps of the full pipeline enabling users to fine-tune their results. Various parameter configurations for a given step can thus be explored without needing to re-run the entire pipeline (Supplementary Fig. 3).

A range of features are included in the plugin, such as the option to delineate a corresponding ROI using the 'Shapes' layer in napari or the option to choose between rigid (CPD) and nonlinear registration (BCPD). Parameters such as the FM segmentation settings, point cloud sampling density and the maximum number of iterations for the point cloud registration can be tuned. Parameters can be saved and reused to ensure reproducibility. Overlays can be directly exported from napari, as well as the initial and transformed point clouds, for quality control purposes. Intermediate outputs such as segmentation masks and sampled point clouds can be directly visualized in the napari viewer to aid troubleshooting. It is also possible to inject intermediate results from external sources, for example, an FM segmentation method from a different plugin, or a pre-existing EM segmentation mask. The resulting 'Labels' layer can then be set as an input to subsequent steps in CLEM-Reg. The results, usability and user-friendliness of the plugin were assessed on three vCLEM datasets (EMPIAR-10819, EMPIAR-11537 and EMPIAR-11666).

### Identifying TGN46-positive transport carriers with CLEM-Reg

To validate CLEM-Reg, a rare dynamic cellular process involving submicron organelles and transport carriers was studied. Despite being primarily localized to the TGN, a small subset of TGN46 (1–5%) rapidly cycles between the TGN and plasma membrane. The TGN46 exits the TGN in 'CARriers of the TGN to the cell Surface' (CARTS), which play a role in transport of plasma membrane proteins (for example, desmoglein-I, a key component of desmosomes) and secretion (for example lysozyme C, pancreatic adenocarcinoma upregulated factor) and recycles back to the TGN via endosomes[33–36]. Leveraging the complementary information derived from vCLEM, unambiguous identification of the subset of endosomes and transport carriers engaged in the process of trafficking TGN46 between TGN and plasma membrane was demonstrated on two datasets (EMPIAR-10819 and EMPIAR-11537) with CLEM-Reg (Fig. 6a,b). Labeling accuracy of endosomes with GFP-TGN46 signal was quantitatively assessed on EMPIAR-11537 (Extended Data Fig. 6) with further quantifications demonstrating equivalent performance on conventional resolution confocal microscopy data (Supplementary Fig. 2c,d) shown in Extended Data Fig. 10.

By utilizing off-target landmarks (in this case, mitochondria) to derive the warping matrix, precise and accurate overlays of fluorescence to small target organelles were achieved, while minimizing subjective errors that result from aligning fluorescence directly to presumed target structures. Here, the overlays facilitated the identification of different morphological populations of TGN46-positive CARTS in different cellular locations; small vesicular structures tended to localize in the periphery, whereas complex membrane and vesicular structures were closer to the perinuclear region (Supplementary Figs. 4 and 5). Such functional labeling of organelles with submicron precision has the potential to provide vital mechanistic information for a range of biological processes.

### Discussion

This study introduces CLEM-Reg, an automated and unbiased end-to-end vCLEM registration framework, implemented as a dedicated plugin for napari. The proposed registration approach relies on extracting landmarks using classical image processing and deep learning. After segmenting internal landmarks in FM and EM, point clouds are sampled and preprocessed to obtain a memory-efficient, modality-agnostic representation. Point cloud registration finds a mapping between the two image volumes, with which the FM acquisition is warped onto the EM volume to obtain a final vCLEM overlay. CLEM-Reg drastically reduces registration time to a few minutes and achieves near expert-level performance on three benchmark vCLEM datasets.

CLEM-Reg's potential in driving new biological insights is also demonstrated. Correlation of small punctate structures, like the transport carriers and endosomes studied herein, is challenging due to the lack of characteristic morphologies. Thus, accurate correlation requires precise and unbiased landmark-based alignment of off-target structures, which CLEM-Reg automatically delivers across the whole cell volume. Wakana et al. studied TGN46 carriers in two dimensions using permeabilized cells[36]. CLEM-Reg, however, enables identification of TGN46 carriers in 3D across whole cells with intact membranes. Future work could build on these results by studying morphological differences of transport carriers in their native state between healthy and pathological cells. Since all registered datasets have been publicly deposited, others interested in TGN46 trafficking can mine these data to more completely study instances of TGN46-positive CARTS. Overall, these results highlight a broader concept for underpinning structure–function studies with vCLEM.

Nevertheless, certain limitations remain with regard to the automated organelle segmentation in EM, which adversely impacts the alignment performance when more than 40% of mitochondria are missed during the segmentation step (Extended Data Fig. 1a). Registration performance with CLEM-Reg is also more sensitive to the loss of

**a** TGN46 overlay on FIB-SEM data (EMPIAR-10819)

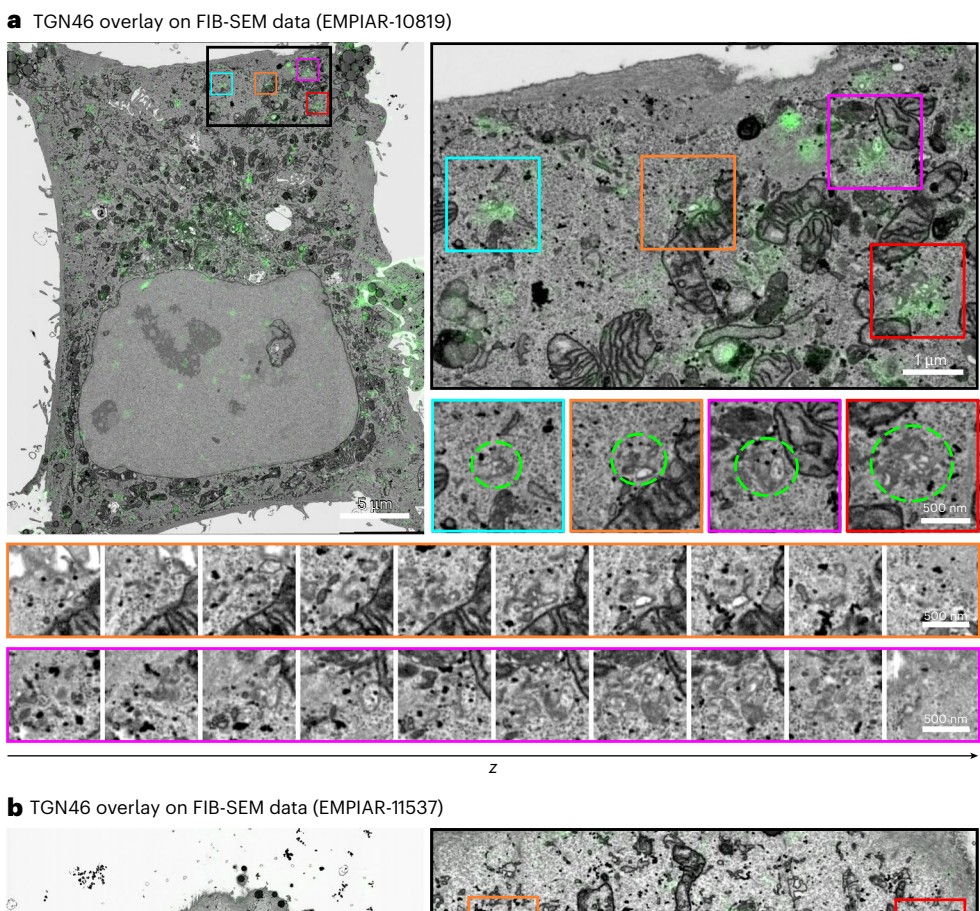

**b** TGN46 overlay on FIB-SEM data (EMPIAR-11537)

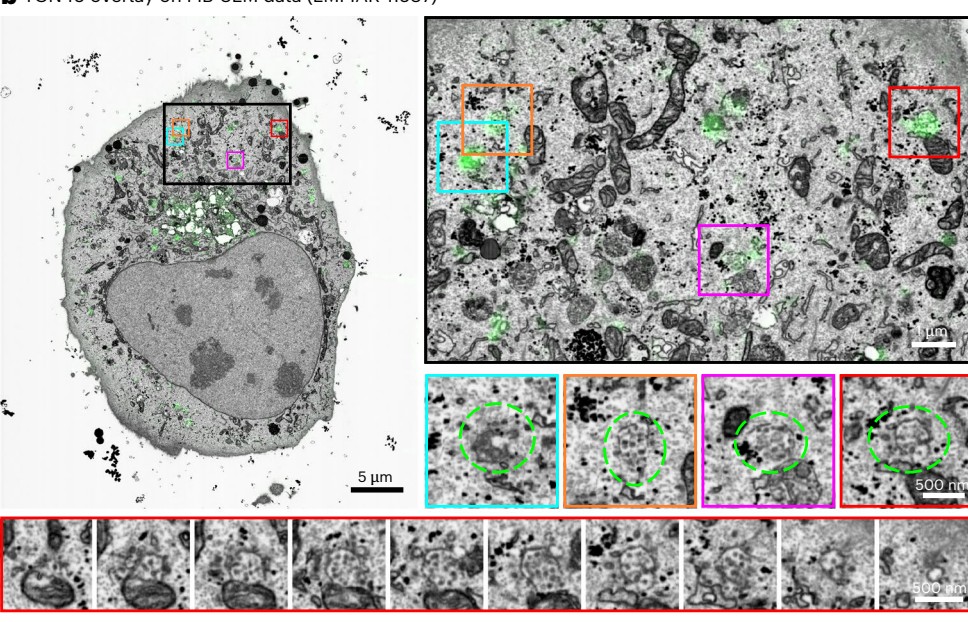

**Fig. 6 | Identification of GFP-TGN46-positive endosomes and transport carriers. a,b**, In addition to the expected TGN localization of GFP-TGN46, GFP-TGN46-positive endosomes and transport carriers were identified in EM guided by overlays obtained with CLEM-Reg. Structures of interest are circled in green and corresponding montages showing entire endosomes or transport carriers are displayed. Images in montages are shown with a spacing of 40 nm (orange) or 50 nm (magenta and red) in *z*. Full montages with 10 nm spacing in *z* can be found in Supplementary Figs. 4 and 5.

peripheral landmarks (Extended Data Fig. 1b) which is consistent with previous work by Paul-Gilloteaux et al.[15]. While MitoNet was shown to perform well on the three vCLEM datasets shown in this study, this cannot be guaranteed for all vEM datasets. For instance, to segment mitochondria in SBF-SEM, preprocessing with contrast-limited adaptive histogram equalization was required (Methods). This is likely due to the imbalanced dataset used to train MitoNet with 75.6% of data acquired on FIB-SEM and only 5.2% on SBF-SEM microscopes[25].

One approach to address this is to make use of transfer learning, a deep-learning method in which already trained models are briefly retrained on a smaller dataset to improve performance. While training deep neural networks previously required access to GPUs and fluency in programming languages such as Python, open-source projects such as ZeroCostDL4Mic[40], DeepImageJ[41], Imjoy[42] or DL4MicEverywhere[43] are rapidly removing these barriers, enabling GUI-driven interaction with a range of deep-learning architectures. Open-source

and community-driven model libraries such as the Bioimage Model Zoo[44], which allow easy sharing of pretrained deep-learning models, are another important resource.

CLEM-Reg includes the option to perform nonlinear alignment of volumes using BCPD and 3D thin-plate-spline warping; however, rigid registration applied to a FIB-SEM benchmark dataset outperforms nonlinear registration (Extended Data Fig. 4) and runs orders of magnitude faster (2.17 min versus 205 min to obtain overlays for all channels, including mitochondria segmentation; Methods). While nonrigid registration is often thought to deliver superior alignment performance, both the results obtained here and previous work[14,15] do not support this assumption. A possible explanation for this counterintuitive finding is that better performance can be achieved by restricting the degrees of freedom, as the point cloud data from the two modalities are inherently noisy and do not perfectly match. Thus, as noise increases due to factors such as segmentation errors, non-isotropic pixel sizes in the FM volume or point sampling, the nonlinear registration approach is more prone to converging to nonoptimal solutions due to the larger parameter space. Rigid registration restricts the degrees of freedom and thus the parameter space, which facilitates convergence toward an optimal solution.

The size of bioimage datasets can also cause challenges for automated alignment. While the time required to register point clouds follows a power law (Extended Data Fig. 3), segmenting and warping image volumes scales cubically. Thus, downsampling of the EM dataset is generally required before execution. Chunked data handling with next-generation file formats (NGFFs)[45] in combination with tools such as Dask[46] could allow users to run the CLEM-Reg on full-resolution data without encountering memory issues. Different use-cases will likely require different resolutions for processing[47] (Extended Data Fig. 2). For instance, segmentation of larger organelles such as mitochondria and nuclei can, in principle, be accurately performed on substantially downscaled data, whereas smaller organelles require retention of higher-resolution information. An appealing feature of NGFFs is that they store data in an image pyramid of multiple resolutions, allowing the selection of appropriate resolutions to accurately and efficiently perform each task without requiring multiple copies. Additionally, CLEM-Reg relies heavily on Scipy[48] and scikit-image[49] for tasks such as FM segmentation and point sampling of the FM and EM segmentation; however, these packages do not natively provide GPU acceleration. As such, future work to incorporate GPU acceleration into the CLEM-Reg workflow using Python libraries such as Cupy[50] or clEsperanto[51] could reduce processing time. This optimization could considerably accelerate multiple steps within the CLEM-Reg workflow.

While CLEM-Reg's ability to register vEM techniques that image the block face is demonstrated, other vEM methods such as array tomography[52] and serial section transmission electron microscopy (ssTEM)[53] could be used given a sufficiently accurate section-to-section alignment; however, robust alignment of consecutive EM sections is challenging, due to the nonlinear deformations associated with manual sectioning and imaging, in particular by transmission electron microscopy (TEM). Currently, each section is finely aligned to the previous section using manually placed landmarks on adjacent sections. This process can introduce bias and is extremely lengthy, requiring days of an expert's time, and is, therefore, beyond the scope of this paper. We speculate that methods based on registering segmented landmarks, such as CLEM-Reg, could form part of an iterative pipeline to improve this section-to-section alignment by providing additional information about the quality of alignment.

While CLEM-Reg has been demonstrated to register single-cell scale data with mitochondria segmentations, the core of the alignment workflow is agnostic to the segmented landmarks. As such, the method could be used at different scales or across different modalities as long as certain conditions are met. These are the existence of a segmentation algorithm for the same structure in both modalities, and that those structures are numerous and distributed throughout the volume to be aligned. For example, nucleus segmentation algorithms in EM and FM could be used to derive point clouds to register tissue-scale volumes for vCLEM, or nucleus segmentations in X-ray microscopy could be used to align to EM or FM (or both). With the rapid development of artificial intelligence-based segmentation methods[54], the range of suitable target structures will likely continue to grow. These developments hold the promise of broadening the applicability of CLEM-Reg to other multimodal registration tasks beyond vCLEM.

## Online content

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

## Methods

### Cell model

HeLa cells were obtained from the Cell Services Science Technology Platform at The Francis Crick Institute and originated from the American Type Culture Collection (ATCC; CCL-2).

### CLEM data acquisition

HeLa cells were maintained in Dulbecco's modified Eagle medium (Gibco) supplemented with 10% fetal bovine serum at 37 °C and 5% $CO_2$. A total of 150,000 cells were seeded in 35-mm photo-etched glass-bottom dishes (MaTtek Corp). At 24 h, cells were transfected with 0.1 μg of GFP-TGN46 construct per dish, using Lipofectamine LTX and PLUS reagent (Invitrogen) in Opti-MEM medium (Gibco) as recommended by the manufacturer. Then, 300 μl of transfection mix were added to 2 ml antibiotic-free medium and incubated overnight. At 36 h, cells were stained with LysoTracker red (100 nM) or Wheat Germ Agglutinin (WGA) Alexa Fluor 594 (5 μg ml$^{-1}$) and MitoTracker deep red FM (200 nM) for 10 min in HBSS (WGA) or 30 min in DMEM (LysoTracker and MitoTracker). Hoechst 33342 (1 μg μl$^{-1}$) was added in the last 5 min of the tracker incubation. All probes were from Molecular Probes (Thermo Fisher Scientific). Cells were then washed three times with 0.1 M phosphate buffer, pH 7.4 and fixed with 4% (v/v) formaldehyde (Taab Laboratory Equipment) in 0.1 M phosphate buffer, pH 7.4 for 15 min. Cells were washed twice and imaged in 0.1 M phosphate buffer on an AxioObserver 7 LSM900 with Airyscan 2 microscope with Zen v.3.1 software (Carl Zeiss). Cells were first mapped with a ×10 objective (NA 0.3) using brightfield microscopy to determine their position on the grid and tile scans were generated. The cells of interest were then imaged at high resolution in Airyscan mode with a ×63 oil objective (NA 1.4). Smart setup was used to set up the imaging conditions. A sequential scan of each channel was used to limit crosstalk and z-stacks were acquired throughout the whole volume of the cells.

The samples were then processed using a Pelco BioWave Pro+ microwave (Ted Pella) following a protocol adapted from the National Centre for Microscopy and Imaging Research protocol (https://emcubed.org/wp-content/uploads/2017/11/ellismans_sbfsem_sampleprep_protocol.pdf). Each step was performed in the Biowave, except for the phosphate buffer and water wash steps, which consisted of two washes on the bench followed by two washes in the Biowave without vacuum (at 250 W for 40 s). All the chemical incubations were performed in the Biowave for 14 min under vacuum in 2-min cycles alternating with/without 100 W power. The SteadyTemp plate was set to 21 °C unless otherwise stated. In brief, the samples were fixed again in 2.5% (v/v) glutaraldehyde (TAAB)/4% (v/v) formaldehyde in 0.1 M phosphate buffer. The cells were then stained with 2% (v/v) osmium tetroxide (TAAB) / 1.5% (v/v) potassium ferricyanide (Sigma), incubated in 1% (w/v) thiocarbohydrazide (Sigma) with SteadyTemp plate set to 40 °C, and further stained with 2% osmium tetroxide in ddH$_2$O (w/v). The cells were then incubated in 1% aqueous uranyl acetate (Agar Scientific) with a SteadyTemp plate set to 40 °C, and then washed in dH$_2$O with SteadyTemp set to 40 °C. Samples were then stained with Walton's lead aspartate with SteadyTemp set to 50 °C and dehydrated in a graded ethanol series (70%, 90% and 100%, twice each), at 250 W for 40 s without vacuum. Exchange into Durcupan ACM resin (Sigma) was performed in 50% resin in ethanol, followed by four pure Durcupan steps, at 250 W for 3 min, with vacuum cycling (on/off at 30-s intervals), before polymerization at 60 °C for 48 h.

Focused ion beam scanning electron microscopy (FIB-SEM) data were collected using a Crossbeam 540 FIB-SEM with Atlas 5 for 3D tomography acquisition (Carl Zeiss). A segment of the resin-embedded cell monolayer containing the cell of interest was trimmed out, and the coverslip was removed using liquid nitrogen before mounting on a standard 12.7-mm SEM stub using silver paint and coating with a 10-nm layer of platinum. The ROI was relocated by briefly imaging through the platinum coating at an accelerating voltage of 10 kV and correlating to previously acquired FM images. On completion of preparation of the target ROI for Atlas-based milling and tracking, images were acquired at 5 nm isotropic resolution throughout the cell of interest, using a 6-μs dwell time. During acquisition, the SEM was operated at an accelerating voltage of 1.5 kV with 1.5 nA current. The EsB detector was used with a grid voltage of 1.2 kV. Ion beam milling was performed at an accelerating voltage of 30 kV and a current of 700 pA.

For SBF-SEM, the block was trimmed to a small trapezoid, excised from the resin block and attached to an SBF-SEM specimen holder using conductive epoxy resin. Before the commencement of an SBF-SEM imaging run, the sample was coated with a 2-nm layer of platinum to further enhance conductivity. SBF-SEM data were collected using a 3View2XP (Gatan) attached to a Sigma VP SEM (Carl Zeiss). Inverted backscattered electron images were acquired through the entire extent of the ROI. For each of the 133 consecutive 50-nm slices needed to image the cell in its whole volume, a low-resolution overview image (horizontal frame width 150.58 μm; pixel size of 75 nm; 2-μs dwell time) and a high-resolution image of the cell of interest (horizontal frame width 63.13 μm; pixel size of 7 nm; using a 3-μs dwell time) were acquired. The SEM was operated in high vacuum with focal charge compensation on. The 30-μm aperture was used, at an accelerating voltage of 2 kV.

### Preprocessing for CLEM registration

Airyscan data were first processed in Zen software using the Airyscan processing tool, which consists in pixel reassignment followed by Wiener filter deconvolution. The settings used were Auto-Filter and Standard strength. The super-resolution Airyscan data was then processed in Zen software using the z-stack alignment tool to correct z-shift. The settings used were the highest quality, translation and linear interpolation, with the MitoTracker channel as a reference. The .czi file was then opened in Fiji[55] and saved as a .tif file. For EMPIAR-11666, the MitoTracker channel was further processed using Yen's thresholding method[56] to remove overexposed punctate artifacts affecting FM segmentation. For the unprocessed data used in Extended Data Figs. 5 and 10, the Airyscan data were first aligned in Zen software using the z-stack alignment tool to correct z-shift, The .czi file was then opened in Fiji. On our system, the raw data size is reduced by grouping the data from the 32 parts of the Airyscan detector into four concentric rings, which appear as four time points in Fiji. Here, the first time point, which corresponds to the central area of the Airyscan detector, was selected and the file saved as a .tif format.

After initial registration with template matching by normalized cross-correlation in Fiji (https://sites.google.com/site/qingzongtseng/template-matching-ij-plugin), the FIB-SEM images (EMPIAR-10819 and EMPIAR-11537) were contrast normalized as required across the entire stack and converted to eight-bit grayscale. To fine-tune image registration, the alignment to the median smoothed template method was applied[57]. The aligned $xy$ 5-nm FIB-SEM stack was opened in Fiji and resliced from top to bottom (to $xz$ orientation) and rotated to roughly match the previously acquired Airyscan data. Finally, the EM volume was binned by a factor of 2 and 4 in $x$, $y$ and $z$ using Fiji resulting in an isotropic voxel size of 10 and 20 nm, respectively.

For the SBF-SEM data (EMPIAR-11666), only minor adjustments in image alignment were needed, carried out using TrakEM2 in Fiji[58]. Finally, the EM volume was binned by a factor of 2 and 4 in $xy$ to obtain a voxel size of 14 × 14 × 50 nm and 28 × 28 × 50 nm, respectively.

### Computational resources

The algorithms described here were primarily developed and tested on a portable machine running on a Linux distribution with the following specifications: 64 GB RAM, Intel Core i7-11850H 2.50 GHz × 16 CPU and GeForce RTX 3080 GPU. An additional workstation with the following specifications was also used for testing: 256 GB RAM, Dual Intel Xeon (14 CPU each) 2.6 GHz, Quadro M600 16 GB GPU.

## FM segmentation with thresholded Laplacian of Gaussian

To ensure the FM pixel values were in the range of 0 and 1, min–max normalization of the following formula was used, where $x$ is the FM volume, $x_{min}$ is the lowest pixel value in the volume, $x_{max}$ is the largest pixel value in the volume and $x_{scaled}$ is the resulting scaled volume $x_{scaled} = \frac{x - x_{min}}{x_{max} - x_{min}}$. In the subsequent step, the scaled FM volume undergoes a LoG filtering process, requiring one tunable parameter $\sigma$. This process involves convolving a one-dimensional difference of Gaussians (DoG) across the $x$, $y$ and $z$ planes of the volume. Marr and Hildreth[59] demonstrated that a reasonable approximation of LoG can be attained by maintaining a ratio of $\frac{\sigma_2}{\sigma_1} = 1.6$ when applying the DoG. A value of $\sigma = 3.0$ was used for EMPIAR-10819 and EMPIAR-11537, thus after applying the LoG ratio, $\sigma_1 = 3.0$ and $\sigma_2 = 4.8$. A dynamic threshold based on a relative user-defined threshold and the mean intensity of a given slice was then applied to each image in the resulting volume to account for inconsistencies arising due to attenuation of the signal at deeper imaging levels. The relative threshold was set to $T = 1.2$ for EMPIAR-10819 and EMPIAR-11537. For EMPIAR-11666, the following parameters were used: $\sigma = 2.2$ and $T = 1.3$. Last, to reduce the number of spurious segmentations, a size-based filter was then applied. This was achieved by using a 3D connected components algorithm from the Python package cc3d (v.3.2.1)[60], which removes components below a user-defined size threshold with a default value set to remove all segmentations with a size below the 5th and above the 95th percentile.

## EM segmentation with MitoNet

The mitochondria within the EM volume were segmented using a pretrained MitoNet deep-learning model[61]. The MitoNet model using its default parameters was applied to EM volumes in EMPIAR-10819 and EMPIAR-11537 without preprocessing, resulting in instance segmentation of the mitochondria. To improve mitochondrial segmentation of EMPIAR-11666, the aligned stack was processed for contrast-limited adaptive histogram equalization[62] using the enhance local contrast (CLAHE) plugin in Fiji, using the following parameters: block size = 127; histogram bins = 256; maximum slope = 2, 3 and 10; mask = "*None*"; fast. Each CLAHE stack (from three maximum slope values) was segmented using MitoNet and the masks were merged by summation before point cloud sampling.

## Point cloud sampling of the EM and FM segmentation

The method employed for producing point clouds from the FM and EM volumes uses the same standardized procedure. First, the FM and EM segmentation masks were resampled to obtain isotropic voxels (EMPIAR-10819 and EMPIAR-11537, 20 nm; EMPIAR-11666, 50 nm). Then, a Canny edge filter[63] with a default value of $\sigma = 1.0$ was applied to each binary segmentation slice. This process ensures that only points on the edge of the outer mitochondrial membrane are randomly sampled. Then, every pixel along the membrane was identified as a potential point to be used within the point cloud, but due to the computationally expensive nature of point cloud registration algorithms, subsequent downsampling was carried out.

## Point cloud downsampling and registration with CPD and BCPD

After sampling, the point clouds were downsampled, binned and filtered by removing statistical outliers to reduce memory requirements and noise. Downsampling of the point clouds was performed by a uniform downsampling function which takes the point cloud and user-defined sampling frequency $1/k$ as the input and samples every $k^{th}$ point in the point cloud, starting with the $0^{th}$ point, then the $k^{th}$ point, then the $k + k^{th}$ point and so on. A value of $k = 30$ was used for all datasets. Following the initial downsampling, a binning step was used which employs a regular voxel grid to create a downsampled point cloud from the input point cloud. There are two primary steps of the binning starting with an initial partitioning of the point cloud into a set of voxels.

Subsequently, each occupied voxel generates exactly one point through the computation of the average of all the points enclosed within it, resulting in a binned version of the original point cloud. A voxel size of $s = 15 \times 15 \times 15$ pixels was used for all datasets. For registration, the Python package probreg (v.0.3.6)[64] was used, as it implements both the CPD[28] and BCPD[27] algorithm with a unified API. For all datasets, rigid CPD with 50 iterations was used to obtain globally registered point clouds of the FM to the EM. The found transformation matrix is then used for the warping of the image volumes. Note that the transformation matrix is computed in pixels, but conversion to physical units can be achieved by an appropriate scaling factor of the translation vector.

## Warping of image volumes

The transformation matrix found from the global point cloud registration was applied to the source image volume using the affine transform implementation provided by SciPy (v.1.10.1)[48], a powerful open-source Python library that provides a range of functions for mathematics, science and engineering. This implementation makes use of inverse projection to interpolate the intensity values of the warped image thus requiring an inverted transformation matrix, which is found with NumPy's (v.1.24.2)[65] linear algebra module. Interpolation of the warped image was achieved with higher-order spline interpolation. Nonlinear image warping was achieved with thin-plate spline deformation which finds a mapping between a set of control points $f : (x, y, z) \rightarrow (x', y', z')$ such that the deformed source points match the target points as closely as possible while ensuring that the image between these control points remains smooth. As there are no open-source implementations for volumetric thin-plate spline warping available for Python, a custom script was implemented based on ref. 38. To reduce computational overhead, the FM volume was nonlinearly warped as eight individual chunks. After warping, FM volumes were resampled to map to the EM space.

## Benchmarking of CLEM-Reg against experts

For expert manual registration, image stacks from FM and EM were manually aligned to each other using the BigWarp plugin of the Fiji framework, with the EM stack set as 'target' and the FM stack as 'moving' dataset. Corresponding points were placed on mitochondria on both datasets and throughout the whole volume of the cell. An affine transformation was applied to the FM data and the transformed dataset was merged with the EM data to produce the final overlay.

Due to the inherent differences in appearance between the two imaging modalities, direct intensity-based quantification of the registration performance is not possible. Therefore, two metrics were introduced to assess registration performance: fluorescent signal overlap to EM structures and centroid distance between fluorescent signal and target structures.

First, target structures (lysosomes in EMPIAR-10819 and EMPIAR-11666 and endosomes in EMPIAR-11537) were manually segmented in 3D in the EM volume in TrakEM2 in Fiji. These lysosomes or endosomes were then cropped with a bounding box encompassing the fluorescent signal from the corresponding manual- or CLEM-Reg-warped FM volumes and then segmented using the Otsu thresholding method[39] to generate the correlating FM segmentation.

Segmentations were transformed from the pixel space to real space by applying appropriate scaling. The intersected volume between fluorescent and EM segmentations was then computed. Centroid distances were obtained by first constructing meshes from segmentations using marching cubes. Then, centroids were determined on meshes derived from fluorescent and EM segmentations and their Euclidean distance calculated. The size of segmented target structures (lysosomes or endosomes) was estimated by computing characteristic length scales $L$ from the volume of the segmented target structures $V$ with $L = \sqrt[3]{V}$.

## Plugin development and deployment

The plugin was developed for Python (v.3.9) and was built with napari (v.0.4.17)[37]. It integrates the CLEM-Reg workflow in the form of a single widget running end-to-end and as a set of individual widgets, each carrying out an individual step (split registration functionality). The EMPIAR-10819 dataset can directly be loaded as sample data from the plugin. To distribute the plugin and make it accessible from the napari interface, the project is packaged using the cookiecutter plugin template provided by napari. The plugin is available for installation via the package installer for Python (pip) in the command line under the project name 'napari-clemreg'. The plugin can also be installed via the built-in plugin menu which lists all plugins currently available on napari-hub. The source code is available as a GitHub repository alongside a detailed project description, installation instructions and user guide. Additionally, a Jupyter notebook is provided to demonstrate headless execution of the workflow without the napari GUI.

## Reporting summary

Further information on research design is available in the Nature Portfolio Reporting Summary linked to this article.

## Data availability

The datasets used in this study have been deposited at EMPIAR and Biostudies. EMPIAR-10819, EM (https://www.ebi.ac.uk/empiar/EMPIAR-10819/) and FM (https://www.ebi.ac.uk/biostudies/bioimages/studies/S-BSST707). EMPIAR-11537, EM (https://www.ebi.ac.uk/empiar/EMPIAR-11537/) and FM (https://www.ebi.ac.uk/biostudies/bioimages/studies/S-BSST1075); and EMPIAR-11666, EM (https://www.ebi.ac.uk/empiar/EMPIAR-11666/) and FM (https://www.ebi.ac.uk/biostudies/bioimages/studies/S-BSST1175).

## Code availability

Code for CLEM-Reg (under MIT license) is available on GitHub (https://github.com/krentzd/napari-clemreg).

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

## Acknowledgements

This work was supported by The Francis Crick Institute which receives its core funding from Cancer Research UK (CC1076, CC1295), the UK Medical Research Council (CC1076, CC1295) and the Wellcome Trust (CC1076, CC1295). D.K. was funded by The Francis Crick Institute, the Pasteur-Paris University International doctoral program, the INCEPTION program (Investissement d'Avenir grant ANR-16-CONV-0005) and a Fondation pour la Recherche Médicale Fin de Thèse grant (FDT202404018132). R.F.L. was supported by a Medical Research Council Skills development fellowship (MR/T027924/1). R.H. is supported by the Gulbenkian Foundation (Fundação Calouste Gulbenkian) and received funding from the European Research Council under the European Union's Horizon 2020 research and innovation program (grant agreement no. 101001332), the European Union through the Horizon Europe program (AI4LIFE project with grant agreement 101057970-AI4LIFE and RT-SuperES project with grant agreement 101099654-RT-SuperES to R.H.), the European Molecular Biology Organization Installation Grant (EMBO-2020-IG4734), the Chan Zuckerberg Initiative Visual Proteomics Grant (vpi-0000000044; https://doi.org/10.37921/743590vtudfp) and a Chan Zuckerberg Initiative Essential Open-Source Software for Science (EOSS6-0000000260). Views and opinions expressed are those of the authors only and do not necessarily reflect those of the European Union. Neither the European Union nor the granting authority can be held responsible for them. M. Russell (Electron Microscopy Science Technology Platform, The Francis Crick Institute and Centre for Ultrastructural Imaging, King's College London) collected preliminary CLEM data used in testing. M.E. was funded by the Cancer Research UK Convergence Science Centre PhD Programme (CANCTA-2024/10007).

## Author contributions

D.K., R.F.L., R.H., L.M.C. and M.L.J. developed the initial idea for this project. D.K. conceived CLEM-Reg and directed this project. D.K., M.E., M.C.D., M.L.J. and L.M.C. wrote the manuscript. D.K. and M.E. developed the source code. C.S. created a Docker file. R.F.L. collected preliminary CLEM data. M.C.D. and C.J.P. collected the benchmark CLEM datasets (M.C.D., FM and SBF-SEM; C.J.P., FIB-SEM). M.C.D. performed manual registrations, segmentations and user testing. D.K., M.E., M.C.D., R.F.L., R.H., L.M.C. and M.L.J. contributed to the revisions of the manuscript.

## Funding

## Competing interests

The authors declare no competing interests.

## Additional information

**Extended data** is available for this paper at https://doi.org/10.1038/s41592-025-02794-0.

**Correspondence and requests for materials** should be addressed to Daniel Krentzel or Martin L. Jones.

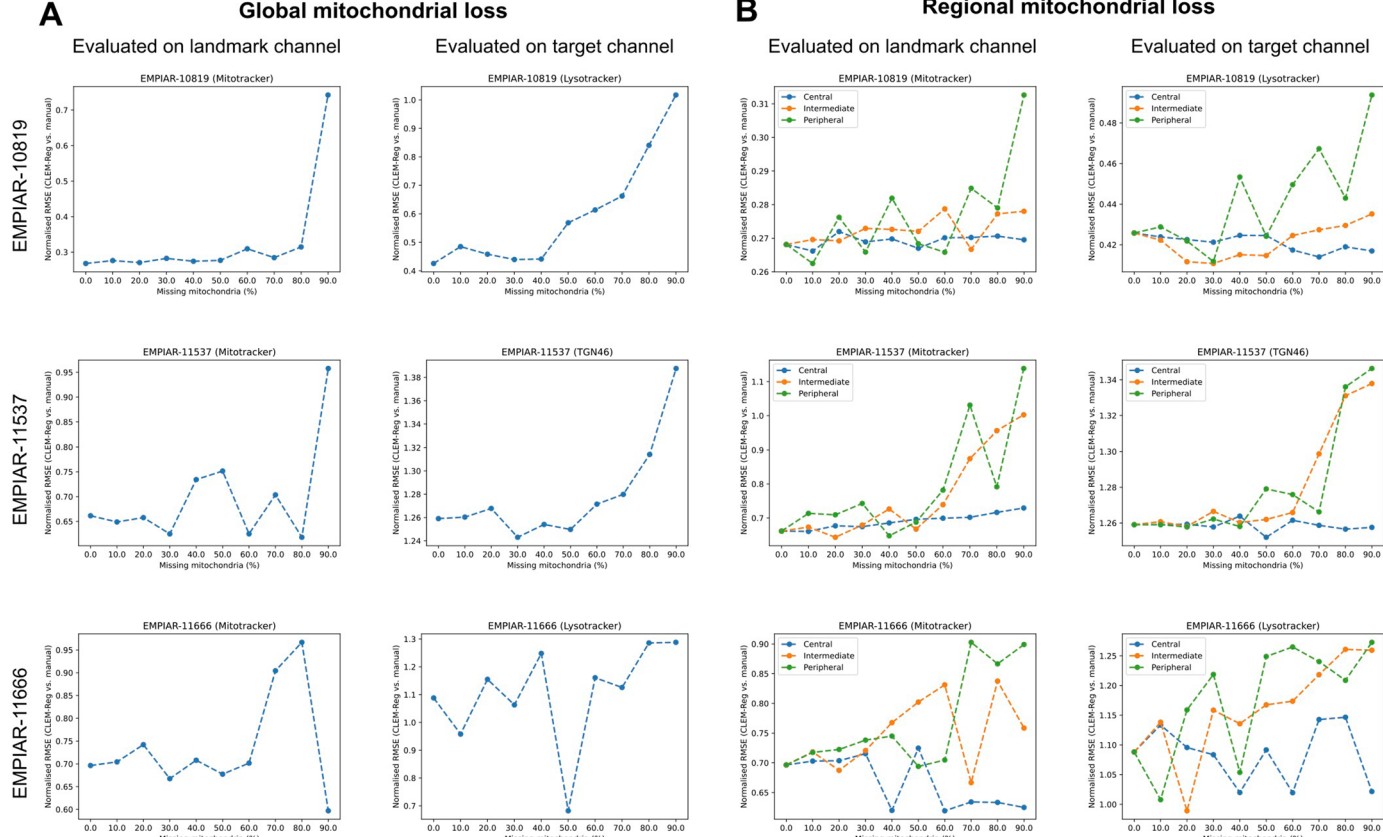

**Extended Data Fig. 1 | Benchmarking registration robustness to missing mitochondria.** The normalized root mean squared error (NRMSE) was computed between manually and automatically obtained overlays with CLEM-Reg as a function of the percent of missing mitochondria in the EM segmentation mask. A fixed voxel size of 10 and sampling frequency of 1/128 was kept for the point cloud generation. (**a**) Mitochondria were randomly removed by identifying mitochondrial instances with connected components and then removing mitochondria with the probability indicated on the x-axis. (**b**) Mitochondria were removed with a probability indicated on the x-axis in three areas with increasing distance from the center of the image volume ('central' in blue, 'intermediate' in orange and 'peripheral' in green).

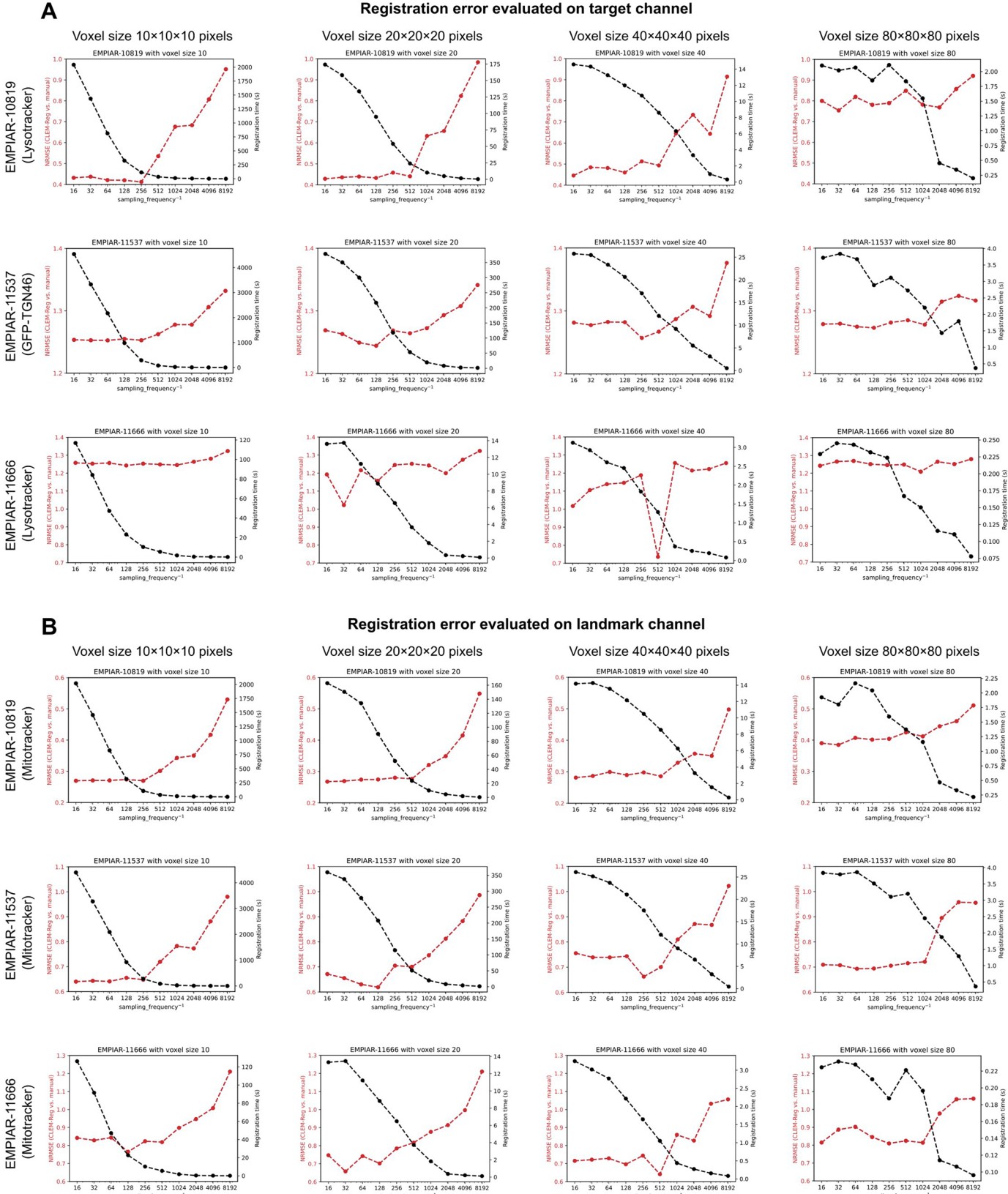

**Extended Data Fig. 2 | Benchmarking registration performance for varying point cloud sampling parameters.** The normalized root mean squared error (NRMSE) was computed between manually and automatically obtained overlays with CLEM-Reg for a decreasing number of sampled points and increasing voxel sizes (pixels) across all three benchmark datasets. (**a**) NRMSE computed on target channels (EMPIAR-10819 and EMPIAR-11666: Lysotracker; EMPIAR-11537: GFP-TGN46) and registration time. (**b**) NRMSE computed on off-target landmark channel (Mitotracker).

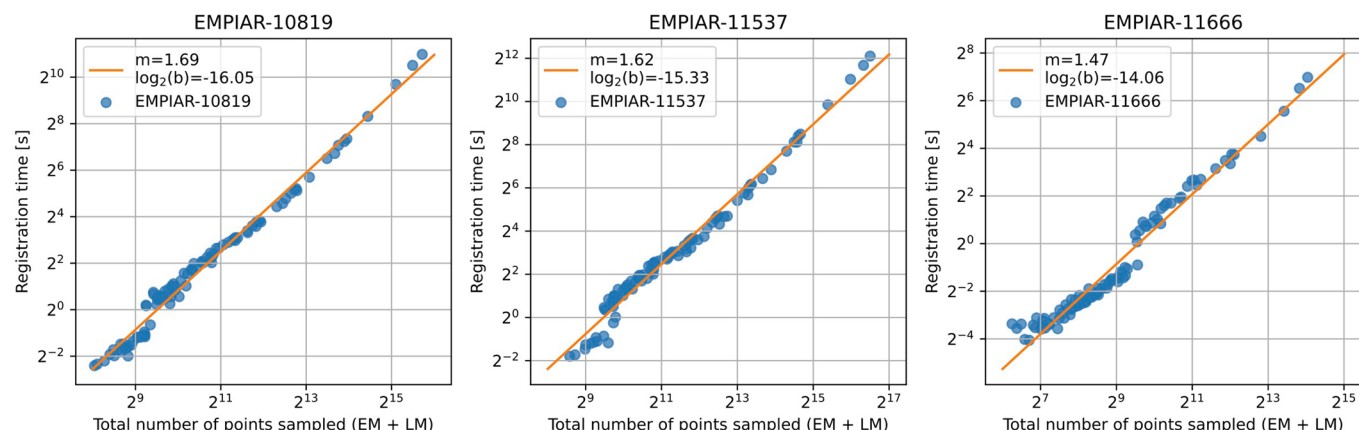

**Extended Data Fig. 3 | Estimating scaling law for rigid CPD registration.**
Empirical data for the number of sampled points and registration time are shown as blue dots and fitted curves in orange are shown on a log-log plot. The parameters governing the power law $y = bx^m$ that relates the number of sampled points to the registration time were estimated with values for the exponent $m$ ranging between 1.47 and 1.69.

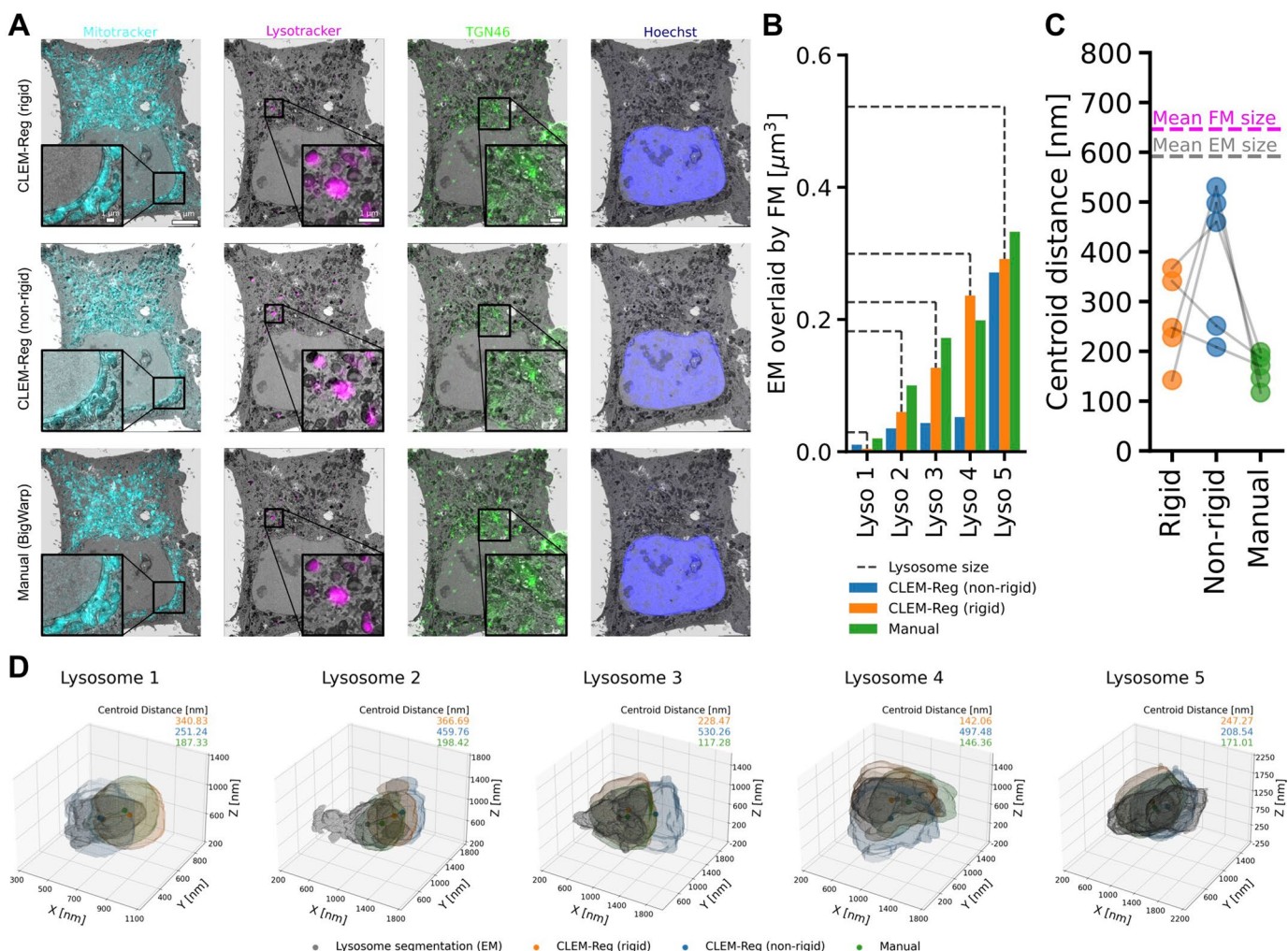

**Extended Data Fig. 4 | Comparing rigid and non-rigid registration results on EMPIAR-10819. (a)** vCLEM overlays showing registration overlays for CLEM-Reg (both rigid and non-rigid) and BigWarp (manual). **(b)** Volume of lysosomes overlaid by fluorescent signal shows that rigid mostly outperforms non-rigid alignment. **(c)** Euclidean distance between centroids segmented in EM and segmented Lysotracker signal shows that in three out of five cases, rigid alignment outperforms non-rigid alignment. **(d)** 3D visualizations of Lysotracker signal overlaid with rigid (orange), non-rigid (blue) and manual alignment (green).

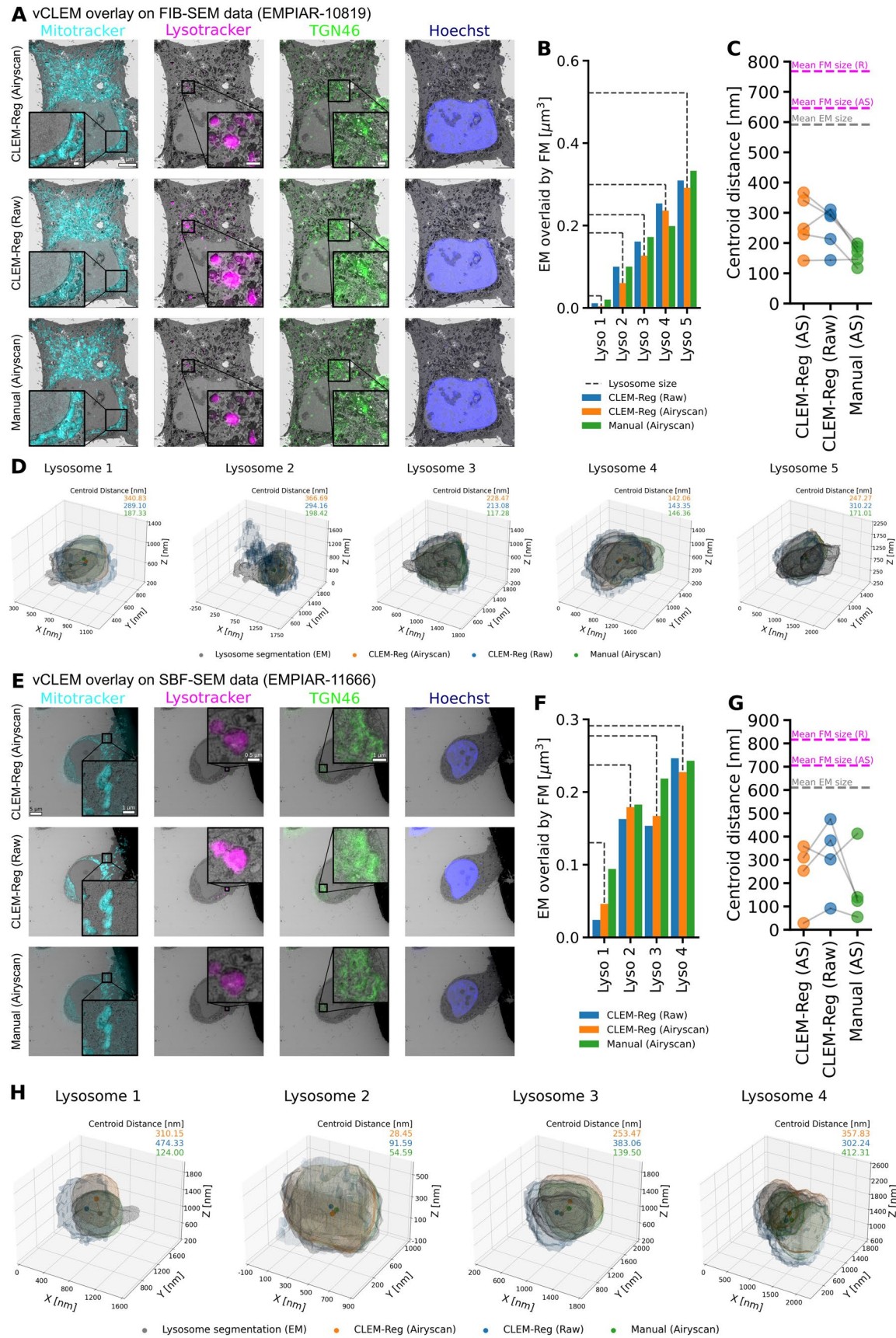

**Extended Data Fig. 5 | See next page for caption.**

**Extended Data Fig. 5 | Comparing alignment results between overlays obtained with CLEM-Reg using raw (unprocessed confocal-like) and processed (super-resolved) Airyscan FM data against overlays obtained by an expert.** (**a**, **e**) Overlays for EMPIAR-10819 (A: FIB-SEM with Mitotracker, Lysotracker, GFP-TGN46 and Hoechst overlaid) and EMPIAR-11666 (I: SBF-SEM with Mitotracker, Lysotracker, GFP-TGN46 and Hoechst overlaid). (**b**,**f**) Volume of lysosomes in EM overlaid by FM signal was computed by intersecting EM with FM segmentations from raw (blue) and Airyscan (orange) FM overlays obtained with CLEM-Reg and Airyscan FM data overlaid by an expert (green). (**c**,**g**) Centroid distances between EM segmentations and segmented Lysotracker signal in FM were computed with Euclidean distance. Airyscan is shown in orange, raw FM data in blue and Airyscan overlaid by an expert with BigWarp in green. Mean size of FM for raw data (indicated by R) and Airyscan data (indicated by AS) and EM segmentations are shown in magenta for FM and gray for EM ($n = 5$ lysosomes in **c** and $n = 4$ lysosomes in **g**). (**d**,**h**) 3D visualizations of lysosome overlays were generated by obtaining meshes from segmentations of EM shown in gray, Lysotracker signal registered using BigWarp (Manual) shown in green, CLEM-Reg on raw data shown in blue and on Airyscan data shown in orange. Corresponding centroid distances are shown next to each lysosome visualization.

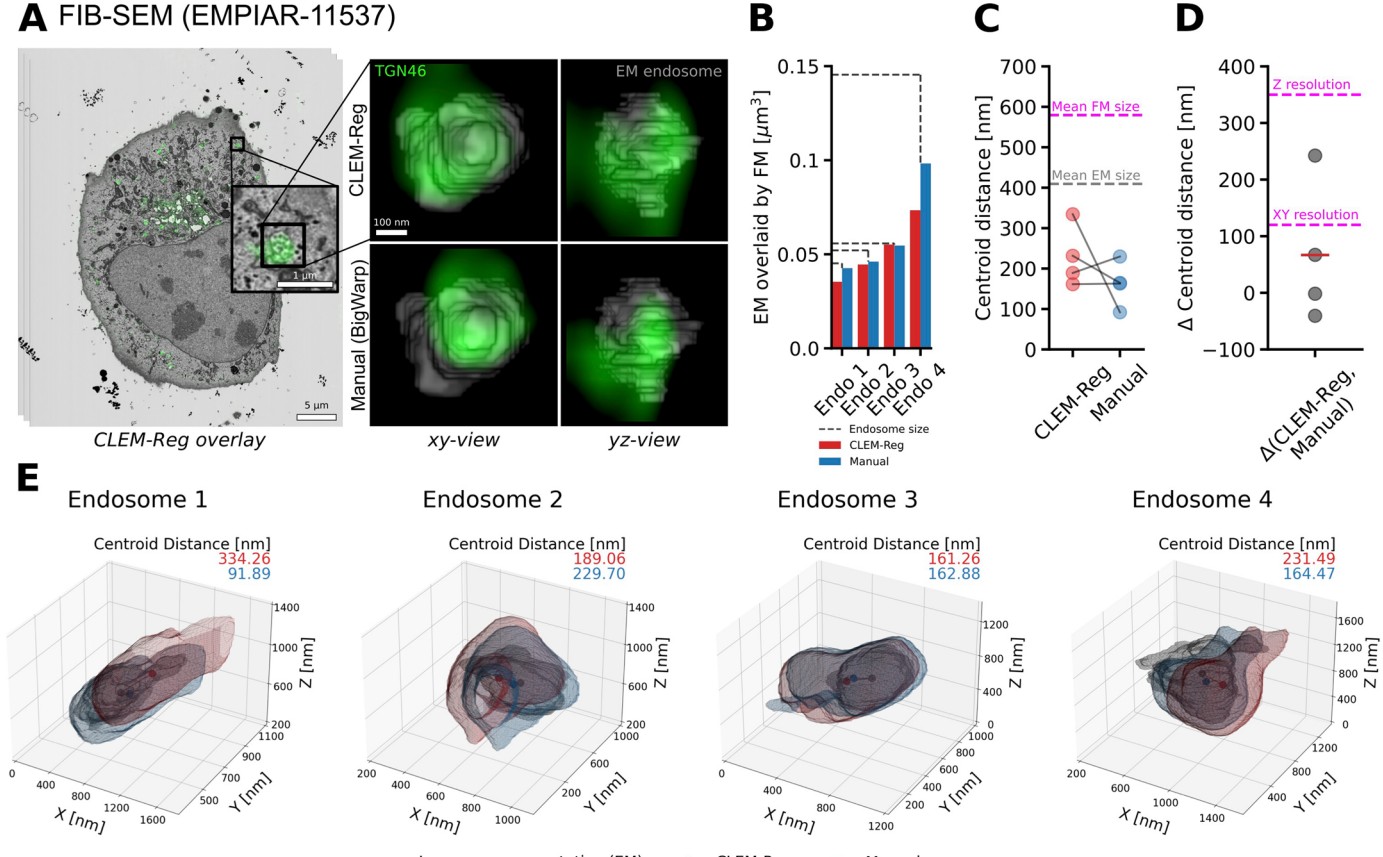

**Extended Data Fig. 6 | Comparing alignment results between CLEM-Reg and experts. (a)** The GFP-TGN46 channel was overlaid to FIB-SEM (EMPIAR-11537) data using mitochondria as off-target landmarks with CLEM-Reg. To quantify registration performance, four endosomes were manually segmented throughout the EM volume. Corresponding segmentations in FM were obtained by segmenting the GFP-TGN46 channel with Otsu's method. **(b)** Volume of endosomes in EM overlaid by FM signal was computed by intersecting EM and FM segmentations. **(c)** Centroid distances between EM segmentations and segmented GFP-TGN46 signal in FM were computed with Euclidean distance. Mean size of FM and EM segmentations are shown in magenta and gray

respectively ($n = 4$ endosomes). **(d)** The difference between GFP-TGN46 signal overlaid manually and with CLEM-Reg was computed from previously found centroid distances. Mean difference in centroid distances is shown with a red horizontal line ($n = 4$ endosomes). The theoretical XY and Z resolution of the fluorescence microscope used is shown in magenta. **(e)** 3D visualizations of endosome overlays were generated by obtaining meshes from segmentations of EM shown in gray, Lysotracker signal registered using BigWarp (Manual) shown in blue and CLEM-Reg shown in red. Visualizations generated with napari and matplotlib.

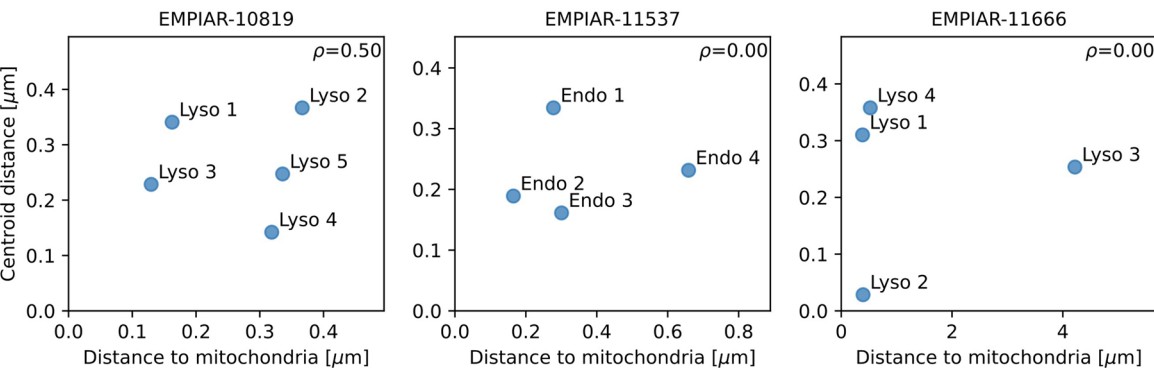

**Extended Data Fig. 7 | Correlating lysosomes and endosomes to the distance to mitochondria.** We obtained distances of lysosomes or endosomes to mitochondria by computing a distance transform on the 3D mitochondria segmentation masks. Distances of lysosomes or endosomes were then obtained by considering the previously obtained distance transform at the centroid of manual lysosome/endosome segmentations. Alignment accuracy was then assessed via centroid distances as shown in Figs. 5c and 5h, as well as Supplementary Fig. 4C. Spearman's correlation was then applied with values shown on the top right hand side corner for each dataset ($n$ = 5 lysosomes for EMPIAR-10819, $n$ = 4 endosomes for EMPIAR-11537 and $n$ = 4 lysosomes for EMPIAR-11666).

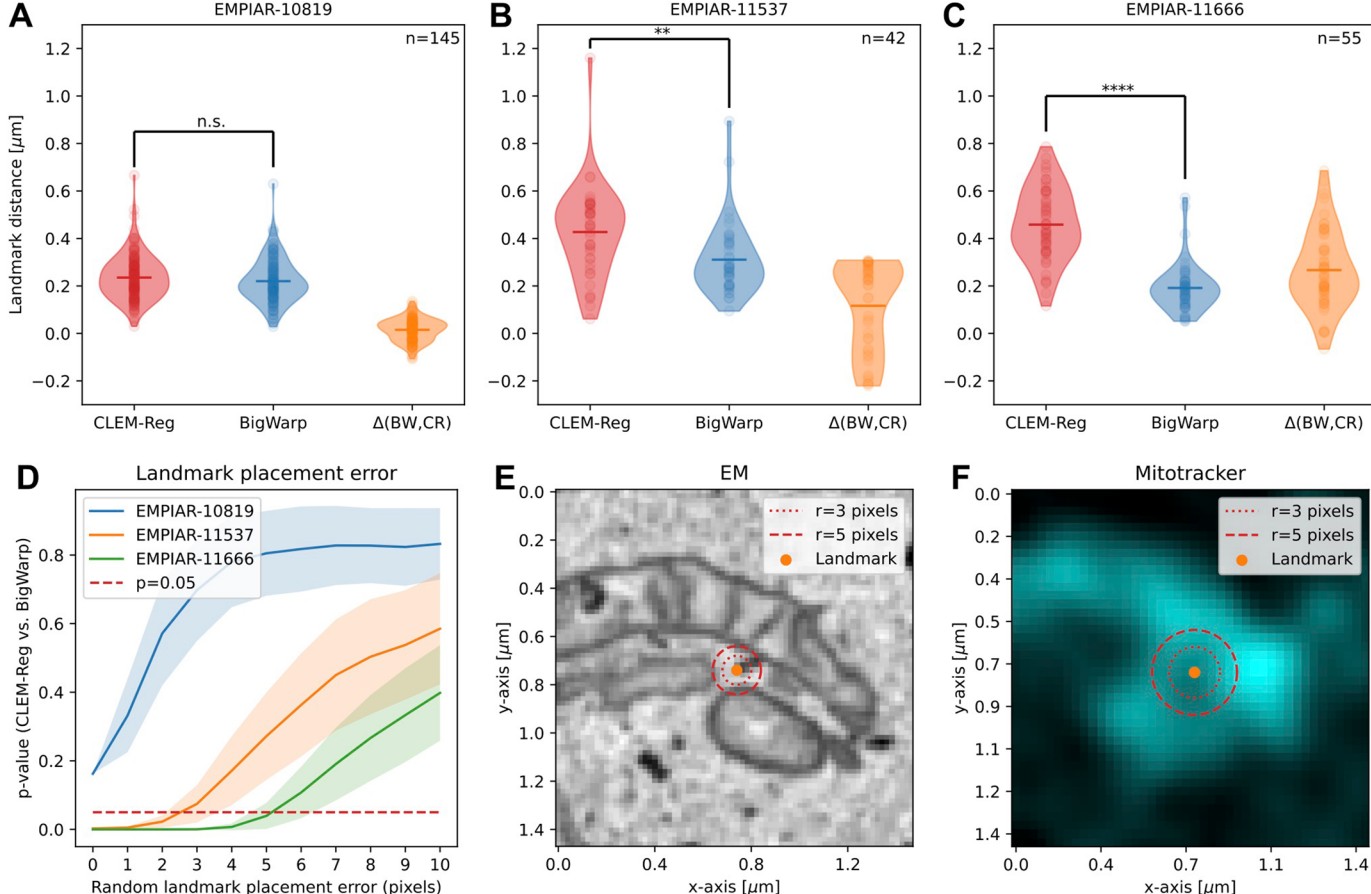

**Extended Data Fig. 8 | Comparing CLEM-Reg against manual alignment with BigWarp on landmarks placed in BigWarp.** (**a**,**b**,**c**) Violin plots showing Euclidean distances computed between landmarks placed in EM and landmarks placed in LM transformed by CLEM-Reg (red) and BigWarp (blue). The difference in distance between landmarks transformed by CLEM-Reg and BigWarp are shown in orange. Mean distances are shown as horizontal bars. Statistical significance was computed with Student's t-test (EMPIAR-10819: $P = 0.16$, EMPIAR-11537: $P = 0.0025$ and EMPIAR-11666: $P = 4.52 \cdot 10^{-19}$) with n.s indicating $P > 0.05$, ** indicating $P < 0.01$ and **** indicating $P < 0.0001$. (**d**) Sensitivity analysis of point placement precision: Landmarks were randomly

perturbed by addition of Gaussian noise with $\mu = 0$ and $\sigma$ corresponding to random landmark placement errors in pixels. Euclidean distances between randomly perturbed landmarks placed in EM and LM transformed by CLEM-Reg and BigWarp were then computed as shown in **a**, **b** and **c** (measurements for each pixel perturbation were repeated $n = 1,000$ times). Lines correspond to mean $P$-values and shaded areas to standard deviations. (**e**,**f**) Representative crops of corresponding landmarks placed in EM (**e**) and LM (F: Mitotracker channel shown) with circles of radius $r = 3$ (red dotted lines) and $r = 5$ (red dashed lines) shown.

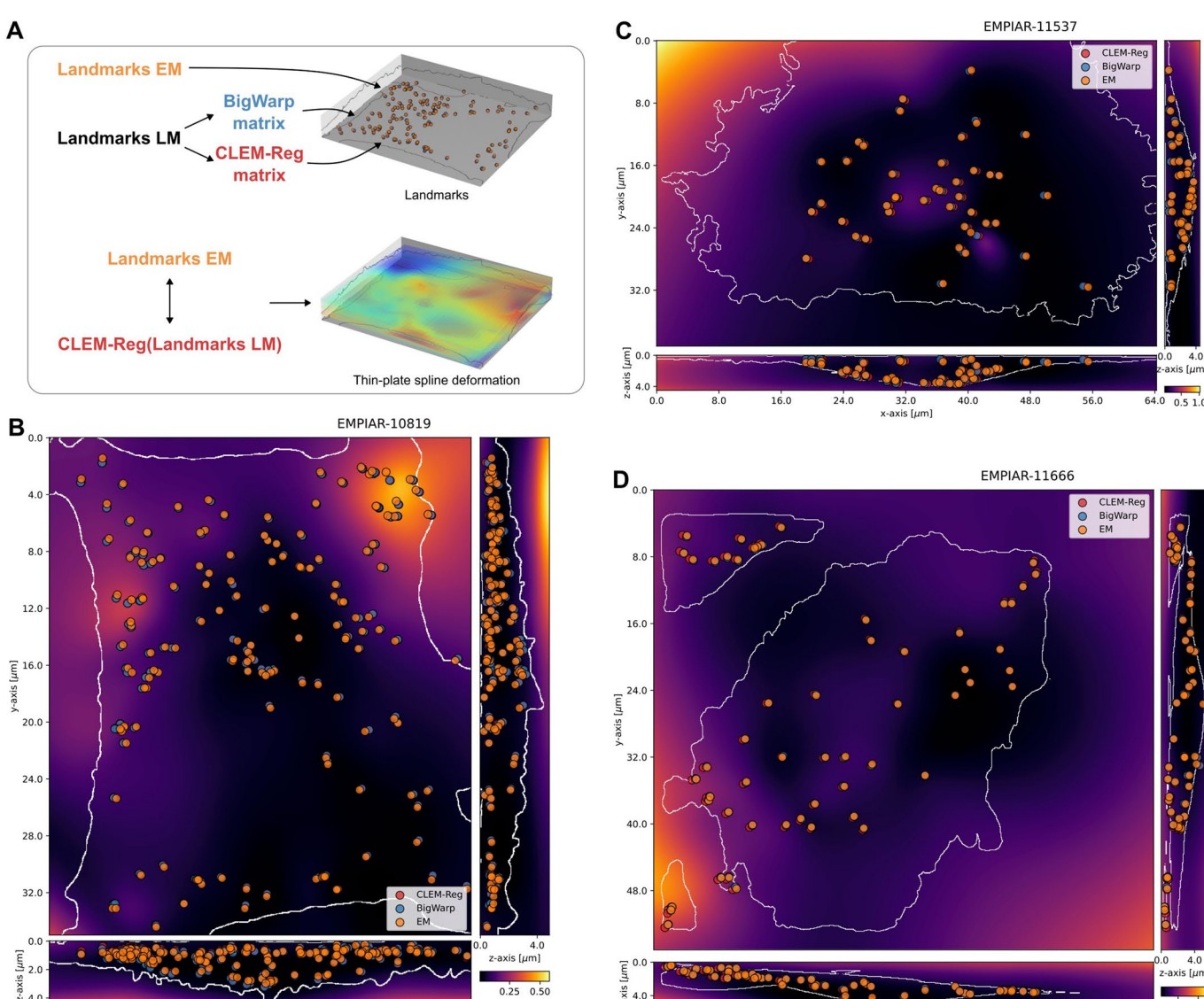

**Extended Data Fig. 9 | Error maps between landmarks placed in EM and LM landmarks transformed with CLEM-Reg.** (**a**) Landmarks manually placed in LM were transformed with the transformation matrix computed by CLEM-Reg (red) and BigWarp (blue). To convert discrete points to a continuous map, a thin-plate spline (TPS) deformation model was computed using landmarks placed in EM (orange) and LM landmarks transformed with the CLEM-Reg matrix (red) as control points. To obtain error maps, the mean squared displacement based on the TPS model was computed across the whole image volume and average projections are shown for EMPIAR-10819 (**b**) EMPIAR-11537 (**c**) and EMPIAR-11666 (**d**) with points corresponding to landmarks placed in EM (orange) and in LM transformed with CLEM-Reg (red) and BigWarp (blue). Maximum projections of cell outlines are shown in white.

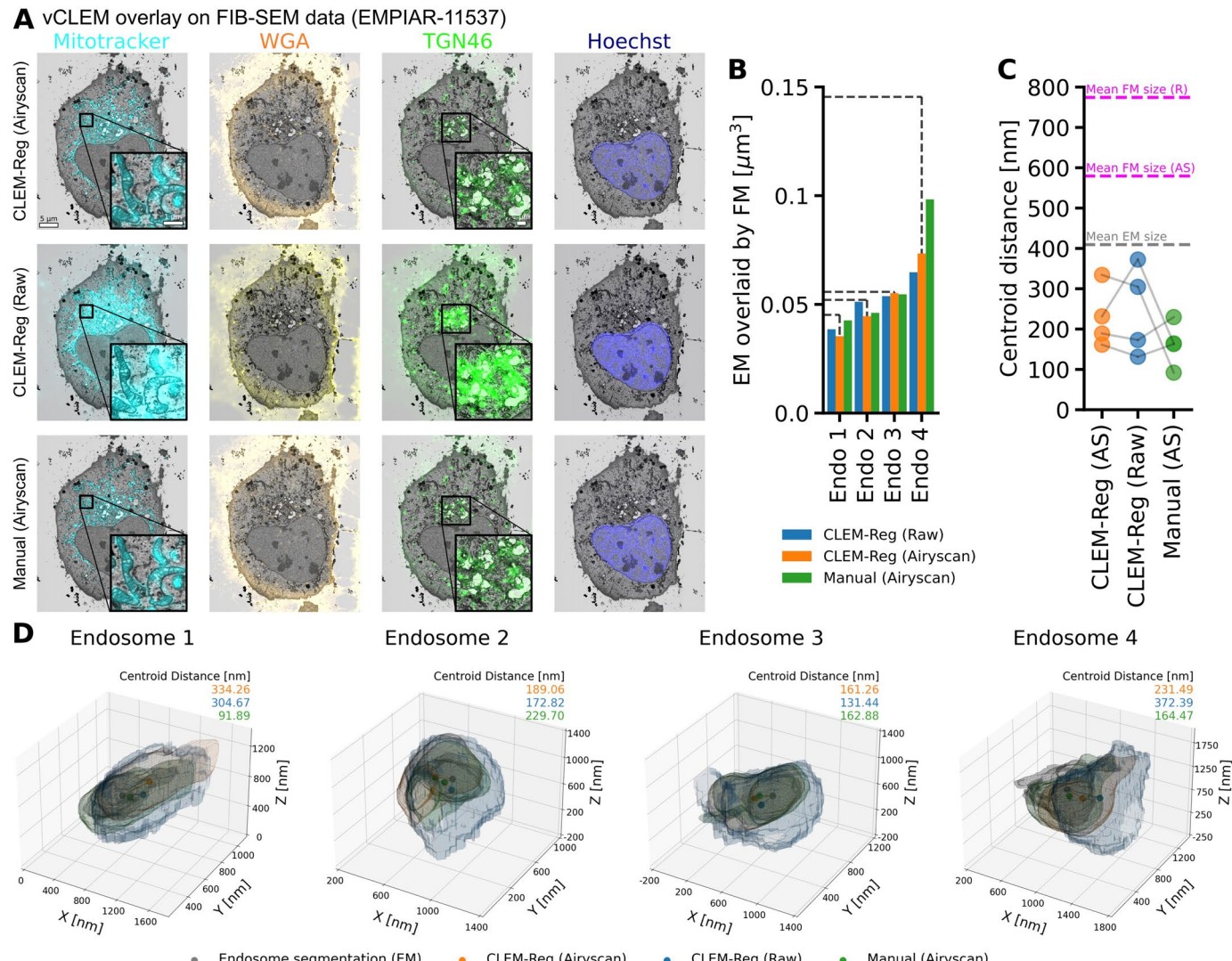

**Extended Data Fig. 10 | Comparing alignment results between overlays obtained with CLEM-Reg using raw (unprocessed, confocal-like) and processed (super-resolved) Airyscan FM data against overlays obtained by an expert.** (**a**) Overlays for EMPIAR-11537 (FIB-SEM with Mitotracker, WGA, GFP-TGN46 and Hoechst overlaid) (**b**) Volume of endosomes in EM overlaid by FM signal was computed by intersecting EM with FM segmentations from raw (blue) and Airyscan (orange) FM overlays obtained with CLEM-Reg and Airyscan FM data overlaid by an expert (green). (**c**) Centroid distances between EM segmentations and segmented GFP-TGN46 signal in FM were computed with Euclidean distance. Airyscan is shown in orange, raw FM data in blue and Airyscan overlaid by an expert in green. Mean size of FM for raw data (indicated by R) and Airyscan data (indicated by AS) and EM segmentations are shown in magenta for FM and gray for EM (*n* = 4 endosomes). (**d**) 3D visualizations of endosome overlays were generated by obtaining meshes from segmentations of EM shown in gray, GFP-TGN46 signal registered using BigWarp (Manual) shown in green, CLEM-Reg on raw data shown in blue and on Airyscan data shown in orange. Corresponding centroid distances are shown next to each endosome visualization.

|  |  |
|---|---|

# Reporting Summary

## Statistics

For all statistical analyses, confirm that the following items are present in the figure legend, table legend, main text, or Methods section.

| n/a | Confirmed | |
|---|---|---|
| ☐ | ☒ | The exact sample size (*n*) for each experimental group/condition, given as a discrete number and unit of measurement |
| ☒ | ☐ | A statement on whether measurements were taken from distinct samples or whether the same sample was measured repeatedly |
| ☐ | ☒ | The statistical test(s) used AND whether they are one- or two-sided<br>*Only common tests should be described solely by name; describe more complex techniques in the Methods section.* |
| ☒ | ☐ | A description of all covariates tested |
| ☒ | ☐ | A description of any assumptions or corrections, such as tests of normality and adjustment for multiple comparisons |
| ☐ | ☒ | A full description of the statistical parameters including central tendency (e.g. means) or other basic estimates (e.g. regression coefficient) AND variation (e.g. standard deviation) or associated estimates of uncertainty (e.g. confidence intervals) |
| ☐ | ☒ | For null hypothesis testing, the test statistic (e.g. *F*, *t*, *r*) with confidence intervals, effect sizes, degrees of freedom and *P* value noted<br>*Give P values as exact values whenever suitable.* |
| ☒ | ☐ | For Bayesian analysis, information on the choice of priors and Markov chain Monte Carlo settings |
| ☒ | ☐ | For hierarchical and complex designs, identification of the appropriate level for tests and full reporting of outcomes |
| ☒ | ☐ | Estimates of effect sizes (e.g. Cohen's *d*, Pearson's *r*), indicating how they were calculated |

*Our web collection on statistics for biologists contains articles on many of the points above.*

## Software and code

Policy information about availability of computer code

| Data collection | Fluorescence Airyscan data acquired with Zen 3.1 software (Carl Zeiss Ltd.). FIB-SEM data acquired with Atlas 5 for 3D tomography acquisition (Carl Zeiss Ltd.). SBF-SEM data acquired with Digital Micrograph (Gatan/Ametek) |
|---|---|
| Data analysis | All code developed is available at https://github.com/krentzd/napari-clemreg<br>Libraries and packages used: Python 3.9,  Numpy 1.24.2, Scipy 1.10.1, Napari 0.4.17, cc3d 3.2.1, probreg 0.3.6 |

For manuscripts utilizing custom algorithms or software that are central to the research but not yet described in published literature, software must be made available to editors and reviewers. We strongly encourage code deposition in a community repository (e.g. GitHub). See the Nature Portfolio guidelines for submitting code & software for further information.

## Data

Policy information about availability of data

All manuscripts must include a data availability statement. This statement should provide the following information, where applicable:
- Accession codes, unique identifiers, or web links for publicly available datasets
- A description of any restrictions on data availability
- For clinical datasets or third party data, please ensure that the statement adheres to our policy

The datasets used in this study have been deposited at EMPIAR and Biostudies. EMPIAR-10819: EM (https://www.ebi.ac.uk/empiar/EMPIAR-10819/) FM (https://www.ebi.ac.uk/biostudies/bioimages/studies/S-BSST707). EMPIAR-11537: EM (https://www.ebi.ac.uk/empiar/EMPIAR-11537/) FM (https://www.ebi.ac.uk/

## Research involving human participants, their data, or biological material

Policy information about studies with human participants or human data. See also policy information about sex, gender (identity/presentation), and sexual orientation and race, ethnicity and racism.

| Reporting on sex and gender | N/A |
|---|---|
| Reporting on race, ethnicity, or other socially relevant groupings | N/A |
| Population characteristics | N/A |
| Recruitment | N/A |
| Ethics oversight | N/A |

Note that full information on the approval of the study protocol must also be provided in the manuscript.

# Field-specific reporting

Please select the one below that is the best fit for your research. If you are not sure, read the appropriate sections before making your selection.

☒ Life sciences       ☐ Behavioural & social sciences       ☐ Ecological, evolutionary & environmental sciences

For a reference copy of the document with all sections, see nature.com/documents/nr-reporting-summary-flat.pdf

# Life sciences study design

All studies must disclose on these points even when the disclosure is negative.

| Sample size | No sample size calculation was performed as no experimental hypotheses were being tested. |
|---|---|
| Data exclusions | No data were excluded |
| Replication | Workflow was executed with a variety of input parameters |
| Randomization | Not relevant to this study, which is a technique/tool development study. |
| Blinding | Not relevant to this study, which is a technique/tool development study. |

# Reporting for specific materials, systems and methods

We require information from authors about some types of materials, experimental systems and methods used in many studies. Here, indicate whether each material, system or method listed is relevant to your study. If you are not sure if a list item applies to your research, read the appropriate section before selecting a response.

## Materials & experimental systems

| n/a | Involved in the study |
|---|---|
| ☒ | ☐ Antibodies |
| ☐ | ☒ Eukaryotic cell lines |
| ☒ | ☐ Palaeontology and archaeology |
| ☒ | ☐ Animals and other organisms |
| ☒ | ☐ Clinical data |
| ☒ | ☐ Dual use research of concern |
| ☒ | ☐ Plants |

## Methods

| n/a | Involved in the study |
|---|---|
| ☒ | ☐ ChIP-seq |
| ☒ | ☐ Flow cytometry |
| ☒ | ☐ MRI-based neuroimaging |

# Eukaryotic cell lines

Policy information about cell lines and Sex and Gender in Research

| | |
|---|---|
| Cell line source(s) | HeLa cells, from Cell Services STP at the Francis Crick Institute |
| Authentication | Not authenticated |
| Mycoplasma contamination | Not tested |
| Commonly misidentified lines<br>(See ICLAC register) | No commonly misidentified cell lines |

# Plants

| | |
|---|---|
| Seed stocks | N/A |
| Novel plant genotypes | N/A |
| Authentication | N/A |

