## [Peer Review File · Nature Methods]

CLEM-Reg: An automated point cloud based registration algorithm for volume correlative light and electron microscopy

Corresponding Author: Dr Martin Jones

A version of this paper was originally rejected for publication by Nature Methods, however that decision was reconsidered after appeal by the authors.

Version 0:

Decision Letter:

9th Jul 2024

Dear Martin,

Thank you for submitting your manuscript entitled "CLEM-Reg: An automated point cloud based registration algorithm for correlative light and volume electron microscopy". We have given the paper our careful consideration but we regret that we cannot publish it in Nature Methods.

It is Nature Methods' policy to decline a substantial proportion of manuscripts without peer-review, so that they may be sent elsewhere without delay. Decisions of this kind are made by the editors when it appears that papers are unlikely to succeed in the competition for limited space.

We read your paper with interest and agree that such tools are timely and will be valuable for volume electron microscopy and CLEM. However, we were concerned about the performance relative to manual registration, and were also concerned that vCLEM is limited to datasets without nonlinear deformations and sectioning artefacts. For these reasons, we could not justify publication in Nature Methods.

Although we cannot offer to publish your manuscript, I suggest that you consider Nature Communications as a suitable venue for this work. To transfer your manuscript, please use our manuscript transfer portal. You will not have to re-supply manuscript metadata and files, unless you wish to make modifications. For more information, please see our [manuscript transfer FAQ](http://www.nature.com/authors/author_resources/transfer_manuscripts.html?WT.mc_id=EMI_NPG_1511_AUTHORTRANSF&WT.ec_id=AUTHOR) page.

Thank you very much for giving us the opportunity to consider your manuscript. I am sorry that we cannot be more positive on this occasion and hope that you will promptly find a more appropriate forum for presenting your work.

Sincerely,
Rita

Rita Strack, Ph.D.
Senior Editor
Nature Methods

For Nature Portfolio general information and news for authors, see <http://npg.nature.com/authors>

Version 1:

Decision Letter:

6th Nov 2024

Dear Martin,

Thank you for your letter asking us to reconsider our decision on your Article, "CLEM-Reg: An automated point cloud based registration algorithm for correlative light and volume electron microscopy". After careful consideration we have decided that we are willing to consider a revised version of your manuscript that is updated as you've described.

- * please underline/highlight any additions to the text or areas with other significant changes to facilitate review of the revised manuscript
- * address the points listed described below to conform to our open science requirements
- * ensure it complies with our general format requirements as set out in our guide to authors at www.nature.com/naturemethods
- * resubmit all the necessary files electronically by using the link below to access your home page

Link Redacted

We hope to receive your revised paper within one month. If you cannot send it within this time, please let us know. In this event, we will still be happy to reconsider your paper at a later date so long as nothing similar has been accepted for publication at Nature Methods or published elsewhere.

OPEN SCIENCE REQUIREMENTS

REPORTING SUMMARY AND EDITORIAL POLICY CHECKLISTS

When revising your manuscript, please submit reporting summary and editorial policy checklists.

DATA AVAILABILITY

CODE AVAILABILITY

Please include a "Code Availability" subsection in the Online Methods which details how your custom code is made available. Only in rare cases (where code is not central to the main conclusions of the paper) is the statement "available upon request" allowed (and reasons should be specified).

For more information on our code sharing policy and requirements, please see:
<https://www.nature.com/nature-research/editorial-policies/reporting-standards#availability-of-computer-code>

MATERIALS AVAILABILITY

ORCID

Nature Methods is committed to improving transparency in authorship. As part of our efforts in this direction, we are now requesting that all authors identified as 'corresponding author' on published papers create and link their Open Researcher and Contributor Identifier (ORCID) with their account on the Manuscript Tracking System (MTS), prior to acceptance. This applies to primary research papers only. ORCID helps the scientific community achieve unambiguous attribution of all scholarly contributions. You can create and link your ORCID from the home page of the MTS by clicking on 'Modify my Springer Nature account'. For more information please visit <http://www.springernature.com/orcid>.

Sincerely,
Rita

Rita Strack, Ph.D.
Senior Editor
Nature Methods

Version 2:

Decision Letter:

22nd Jan 2025

Dear Martin,

Hello and happy new year!

Your Article, "CLEM-Reg: An automated point cloud based registration algorithm for correlative light and volume electron microscopy", has now been seen by three reviewers. As you will see from their comments below, although the reviewers find your work of considerable potential interest, they have raised a number of concerns. We are interested in the possibility of publishing your paper in Nature Methods, but would like to consider your response to these concerns before we reach a final decision on publication. We therefore invite you to revise your manuscript to address these concerns.

In terms of the tool itself, we ask that you focus on addressing the technical concerns and questions, especially regarding rigid vs deformable registration (ref 3) and better quantifying accuracy/error (refs 1 and 2). In terms of the tool itself, we ask that you update documentation as requested by the refs and offer improved user guidance. Regarding the presentation, we ask that you clarify what types of EM data the method is suitable for and how the approach could be made more generally extensible. We also ask that you either explore or discuss the limits of the method in terms of scale.

* include a point-by-point response to the reviewers and to any editorial suggestions

- * please underline/highlight any additions to the text or areas with other significant changes to facilitate review of the revised manuscript
- * address the points listed described below to conform to our open science requirements
- * ensure it complies with our general format requirements as set out in our guide to authors at www.nature.com/naturemethods
- * resubmit all the necessary files electronically by using the link below to access your home page

Link Redacted

We hope to receive your revised paper within three months. If you cannot send it within this time, please let us know. In this event, we will still be happy to reconsider your paper at a later date so long as nothing similar has been accepted for publication at Nature Methods or published elsewhere.

OPEN SCIENCE REQUIREMENTS

REPORTING SUMMARY AND EDITORIAL POLICY CHECKLISTS

EXTENDED DATA FIGURES

DATA AVAILABILITY

All novel DNA and RNA sequencing data, protein sequences, genetic polymorphisms, linked genotype and phenotype data, gene expression data, macromolecular structures, and proteomics data must be deposited in a publicly accessible database, and accession codes and associated hyperlinks must be provided in the "Data Availability" section.

To further increase transparency, we encourage you to provide, in tabular form, the data underlying the graphical representations used in your figures. This is in addition to our data-deposition policy for specific types of experiments and large datasets. For readers, the source data will be made accessible directly from the figure legend. Spreadsheets can be submitted in .xls, .xlsx or .csv formats. Only one (1) file per figure is permitted: thus if there is a multi-paneled figure the source data for each panel should be clearly labeled in the csv/Excel file; alternately the data for a figure can be included in multiple, clearly labeled sheets in an Excel file. File sizes of up to 30 MB are permitted. When submitting source data files with your manuscript please select the Source Data file type and use the Title field in the File Description tab to indicate which figure the source data

pertains to.

Please include a “Data availability” subsection in the Online Methods. This section should inform readers about the availability of the data used to support the conclusions of your study, including accession codes to public repositories, references to source data that may be published alongside the paper, unique identifiers such as URLs to data repository entries, or data set DOIs, and any other statement about data availability. At a minimum, you should include the following statement: “The data that support the findings of this study are available from the corresponding author upon request”, describing which data is available upon request and mentioning any restrictions on availability. If DOIs are provided, please include these in the Reference list (authors, title, publisher (repository name), identifier, year). For more guidance on how to write this section please see: <http://www.nature.com/authors/policies/data/data-availability-statements-data-citations.pdf>

CODE AVAILABILITY

Please include a “Code Availability” subsection in the Online Methods which details how your custom code is made available. Only in rare cases (where code is not central to the main conclusions of the paper) is the statement “available upon request” allowed (and reasons should be specified).

MATERIALS AVAILABILITY

ORCID

Nature Methods is committed to improving transparency in authorship. As part of our efforts in this direction, we are now requesting that all authors identified as ‘corresponding author’ on published papers create and link their Open Researcher and Contributor Identifier (ORCID) with their account on the Manuscript Tracking System (MTS), prior to acceptance. This applies to primary research papers only. ORCID helps the scientific community achieve unambiguous attribution of all scholarly contributions. You can create and link your ORCID from the home page of the MTS by clicking on ‘Modify my Springer Nature account’. For more information please visit www.springernature.com/orcid.

Sincerely,
Rita

Rita Strack, Ph.D.
Senior Editor
Nature Methods

Reviewers' Comments:

Reviewer #1 (Remarks to the Author):

Remarks to the Author:

In the manuscript entitled “CLEM-Reg: An automated point cloud based registration algorithm for correlative light and volume electron microscopy”, Jones et al, introduce a novel image processing plugin that aims at automating multimodal volume imaging dataset registration, with a strong emphasis on correlative light and electron microscopy.

The proposed strategy relies on:

- the identification and automated segmentation of common features in each imaging modality. Here, the authors use the widely dispersed and easily labelled mitochondrial network.
- An efficient 3D segmentation of fluorescent markers

- The most recent advances in deep learning: pre-trained algorithms can be used on local computing capacity, with high performances to segment cell compartments (MitoNet) that, until recently, required labour intensive manual segmentation in the vEM community.
 - The efficient use of a probabilistic registration algorithm to calculate the transformation of one dataset to match the second, based on a large number of points dispersed at the surface of the segmented volumes of interest (the mitochondrial network). To assess the performance of their registration strategy, the authors:
 - rely on a direct comparison with human experts that use an intrinsically different approach of 'a few but acute point pairing', in the absence of existing ground truth dataset in the field.
 - After registration of the dataset, the displacement of the centre of mass of the segmented LM dataset (centroid) is measured in regard to the centre of mass of the EM dataset. The authors use this distance in XY and Z to evaluate the accuracy of the registration.
 - Highlight the importance of using registration landmarks, independent of the labelled structure of interest, to obtain a bias free registration (CARTs localization).
- The complete workflow is compiled into a plugin named CLEM-Reg, accessible in Napari, a python-based image viewer, and seamlessly integrates external plugins for an accessible experience by the end user (often a biologist with limited expertise in image processing or even computer handling).

General impression on the manuscript:

The manuscript is written in a clear style, and the proposed strategy is intelligible to a broad readership. The proposed tool aims at automatising a tedious process that is currently mostly achieved, in 3D, using AMIRA, BigWarp or eC-CLEM. The focus on 3D registration is of utmost importance and in that sense, the software will relieve many scientists from the registration burden.

Personal impression on the strategy:

From the reading perspective, the proposed plugin seems easily accessible and user-friendly, I now have to try installing and testing it. The availability of the test material is a good point, but I should also try it on my own dataset. Registration accuracy is somewhat better when performed by an expert, at the expense of time. Yet, the accuracy discrepancy remains below the diffraction barrier, and therefore, sub-LM resolution accuracy is achieved. This is clearly a strong point towards automation in general.

The possibility to use any landmarks, assuming we can segment it is good and seems easily accessible through the graphical user interface.

Line 107-108: this is a key difference between your registration and a manual one that cannot be understated. Your CPD strategy elegantly addresses the absence of point to point correspondence.

Following a detailed reading, some points came to my attention, and I would like to see them highlighted or discussed:

Installing the software:

Following the proposed installation procedure is not successful. Although it looks like it should be simple, after launching Python 3.8, the command line states:

```
conda create -n clemreg_env python=3.9
File "<stdin>", line 1
conda create -n clemreg_env python=3.9
^
```

SyntaxError: invalid syntax

Please keep in mind that your end-user is a biologist with no expertise in programming. Already installing a plugin in ImageJ must be heavily documented (copy past in the proper folder), so command line is scary.

Transformation model:

The registration comparison is done between an affine registration (manual landmark deposition) and a rigid registration (CLEM-Reg). For an accurate comparison, I would have expected the same transformation model to be proposed. BigWarp (Bogovic et al., 2016) proposes a rigid transformation through its 'similarity' transformation type. Alternatively, an affine transformation form could be proposed in CLEM-Reg. If this is already proposed, then I am sorry if I missed that part. The spline line model is indeed often requested by the community as it seems intuitively more accurate but happens to be counterproductive in many cases. This was already carefully studied by (Paul-Gilloteaux et al., 2017) and (Bogovic et al., 2016) and a direct reference to their work could support this coherent result.

Model orientation:

I understand the benefits of resampling the dataset into a similar orientation for a human expert to achieve good registration: our brain is meant to identify and extrapolate morphological similarities and cannot easily navigate 3D volume with orthogonal orientations. However, is this also necessary for a computer, working on a cloud of points with statistical robustness? Although it seems irrelevant to the material used in this article, it would be a game changer in volume registration of tissue imaged with even more modalities (Karreman et al., 2014). If the authors could discuss (demonstrate?) this point, it would be interesting.

Density of points:

The results demonstrate the benefits of down sampling the number of points on the outer membrane of the segmented structures on the calculation time, without impacting the registration accuracy. Does this relate to an absolute number of points? Or a density per volume? That could be the beginning of a solid benchmarking tool for the future algorithms that the community will build upon CLEM-Reg. eC-CLEM illustrated the limited benefit of adding points beyond a certain number, by monitoring the 'Target Registration Error'. In practice, beyond 15 points in 2D or 25 points in 3D, if they can even be identified, the registration precision is sufficient to accurately localize a structure below the diffraction limit. I would appreciate if the authors could elaborate on that aspect (minimal number, if it even exists).

Imaging resolution:

One key benefit of manual registration is the sub-pixel point placement, that relies on the subjective expert experience to identify the centroid of the homology structure, regardless of the resolution in the three axes for each dataset. I feel that this relates to the discrepancy in accuracy that decays faster with pixel size binning than with points removal, regardless of the

computing time.

Along this line, what would be the impact of a lower resolution by light microscopy, especially in Z, on the final registration accuracy? The dataset presented here is using state-of-the-art light microscopy apparatus, but we can expect also light microscopy datasets of lower resolution (epifluorescence, laser scanning confocal, cryo-LM). Could that cause registration failures? it's an open question for discussion.

A fundamental mathematic question that does not need to be addressed in the manuscript:

Although I doubt my question is mathematically meaningful (unpaired cloud of points), do you think you could extrapolate a registration error prediction, similar to what was done on the 'leave-one-out' (Kukulski et al., 2011) or 'Target Registration Error' (Paul-Gilloteaux et al., 2017)? The use of cloud-point might prevent that as a statistically relevant result. A direct response is sufficient and not need to be discussed in the paper as it might blur the message. Unless the author thinks it is supporting the general purpose of the manuscript.

Minor comments:

Line 83-85: The manual registration is not responsible for introducing bias: the use of the adequate structure for registration is the responsibility of the scientist. Should I have multiple channels to conduct the registration, I would follow your strategy of registering independent labels, even manually. Please rephrase.

156-159: from the text, it seems that you clean up the segmented mask by a simple size base filter. However, in the figure 2A, 3D segmentation (top right panel) looks slightly different from Z=7 (bottom left panel). As if you added a 'dilate' step on the mask. Is it only due to a differential Z selection in the figure? Or is there a missing step in the image processing description?

Line 173: fixed voxel size of 10? What unit? I assume nanometre. The unit is also missing on supp figure 1.

Line 190: the rigid registration outperforms the non-linear alignment... 'data not shown'? I don't need to have it in the paper as it would impair readability, but please mention it's not shown if that's the case.

Line 196: I would not state that it occupies both the centre and the periphery. One could argue about the centre of the dataset or the sample. Across disperse areas of the cells seems accurate enough.

Line 222-223: where are located the more accurately registered lysosomes? Are they at the periphery? at the centre? What could account for that? The transformation model (rigid/affine) or the distance to the centre of the dataset? Or is it just random? This should be discussed.

Line 337: 'those structures are numerous and distributed...' I already commented on the number of points required (absolute number or density threshold), but I am also interested in the distribution parameters. eC-CLEM elaborated significantly on this aspect, illustrating the importance of homogeneous distribution. I assume this is comparable. A 2D projection of the manual landmark would be appreciated.

Line 459: voxel-size of s=15 what? Unit?

Comment on the figures:

Figure 1C: could the channels be converted to black and white for better visibility? The green channel is barely visible in the received document. I would also suggest increasing the dpi of the figure, so the lines of the GUI become readable. On a printed version, it is not readable at all. Furthermore, in the user manual of the software, I would strongly recommend giving a detailed description for each parameter independently: an 'interpolation order' for 'image warping' does not mean much to most biologist.

Supplementary figure 1: A- on target channel (title) or off-target channel (legend)? Same question for B. A and B: Voxel size 10, 20, 40, 80 (what unit?).

Supplementary note line 19-20: I'm not sure to understand the binning part. If many points are withing the same binning area, they become 1, regardless of the number? Meaning that anything ranging from 0 to X points is turning into 0 or 1 whiting that voxel? Could you elaborate more precisely in the text?

Supplementary figure 3: same comment as for figure 1C. LM in black and white, and larger screenshots of the split workflow would help readability (at least in the user manual).

Supplementary figure 7A: Robustness to miss-labelling is an important result when it comes to 40% loss! I understand that the figure gets in supp data, but I would recommend empathizing it in the discussion. Personal comment: this is not the scope of the paper, but this might be used by MitoNet to improve training of their software (especially on SBF datasets) in a feedback loop.

Supplementary figure 7B: observing that loss of peripheral landmarks is impacting more significantly the registration accuracy than central elements is coherent with already published work, see previous comment (Paul-Gilloteaux et al., 2017).

Supporting materials

Could the author provide video tutorials, similar to what is available on BigWarp or eC-CLEM? This would be certainly helpful to the end-user and support fast spread of the tool to the community. Installing the software and third-party elements; running the software natively, importing external landmarks etc.

Reviewer #1 (Remarks on code availability):

Following the proposed installation procedure is not successful. Although it looks like it should be simple, after launching Python 3.8, the command line states:

```
conda create -n clemreg_env python=3.9
```

```
File "<stdin>", line 1
```

```
conda create -n clemreg_env python=3.9
```

```
^
```

```
SyntaxError: invalid syntax
```

Please keep in mind that your end-user is a biologist with no expertise in programming. Already installing a plugin in ImageJ must be heavily documented (copy past in the proper folder), so command line is scary.

Reviewer #2 (Remarks to the Author):

Summary

The authors present an automatic image registration method (CLEM-Reg) for correlative light and volume electron microscopy (vCLEM).

CLEM-Reg identifies common landmarks in the two modalities using automatic mitochondria segmentation from EM and requires a fluorescence microscopy image also containing mitochondria. CLEM-Reg then applies the coherent point drift algorithm to the extracted landmarks to find the transformation which is then applied to the light image data. The authors evaluate and validate CLEM-Reg by measuring the alignment of an independent structure (lysosomes) and comparing its results to manual registration. CLEM-Reg is open source, with code available on github, and is distributed as a plugin for napari.

The presentation is clear and well-motivated, though I have some minor questions. Overall, the authors have convinced me that the method has utility for researchers. The evaluation is sound, though I have some larger concerns that I describe below, along with some ideas that I hope are constructive.

Novelty and significance

While some of the general ideas employed by this work have been done before, the specifics are novel, to my knowledge. Deep learning has been used for CLEM registration in general as in citation (8). Deep learning has been used to generate landmarks for (Ekvall), and segmentation has been used for registration of light and electron microscopy (Zheng). This is the first work that I have seen that uses landmarks derived from a segmentation algorithm for CLEM registration.

The authors correctly say in the discussion that the general idea presented could be applied to a wide range of multi-modal registration. However, the method as presented uses segmentation of mitochondria as a source of landmarks for registration. As such the method is specific to cell-biological samples, or would need to be significantly modified to be applied more broadly.

Evaluation

Evaluating registration is challenging and I appreciate the authors' efforts to do so in an unbiased way. The qualitative comparisons are useful, but a concern I have is the relatively few data points used in the quantitative comparison. As I understand, nine lysosome segmentations (five in EMPIAR-10819, and four in EMPIAR-11666), and four endosome segmentations in EMPIAR-11537 are used in quantification. It's hard to be especially confident in the results when the number of data points is relatively small.

Ideas:

Since an expert has already clicked landmark pairs for these data as part of this evaluation, the authors could use the transformation estimated by clem-reg to transform the moving landmark placed by the expert and measure its distance to the corresponding point placed manually. This would give a measure of agreement with the manual method for every placed landmark.

Even further, it is possible to sample points in your moving image and 1) transform them with the clem-reg transformation 2) transform them with the manual transformation 3) compute their distance, to create a dense "heat map" of errors over space. This may or may not be that informative, but it should be straightforward to try.

Related questions:

How many landmarks were placed by the expert using BigWarp? What kind of transformation was used in BigWarp? (Figure 4's caption suggests affine - was that 12 degrees of freedom?). Please clarify this. Related - does the transformation for the "rigid" CPD have 9 degrees-of-freedom? If there is a difference, then it could affect how to interpret this evaluation.

Using mitos as a fiducial makes sense, especially given the success of mito-net. I wonder about the accuracy of registration far from mitochondria though. Do the authors know if results are similarly good throughout the volume - especially far from mitos? It would be interesting to know if this has been tested. Or, if there is uncertainty and the authors agree that there is a risk that results are worse far from mitos, then it would be good to make researchers working on such structures aware of this. From supplementary Figure 2, it appears that the evaluated lysosomes are all close to mitochondria, but it is difficult to know. Supp Fig 7B may be informative when it comes to this question, given the "central / intermediate / peripheral" labels. Qualitatively, it appears that removing intermediate and peripheral mitos has a larger effect. I would be interested to hear the authors' interpretation of this, if they have ideas.

Minor Questions

line 169:

"extraction of coordinate of pixels"

Are the transformations computed in pixel or physical units?

line 189:

do they authors have an idea as to why rigid outperformed non-linear alignment? This would be an interesting

line 199:

does the reported time include the time needed for segmentation of mitos?

Citations

Using Deep learning and landmarks:

Ekvall et al. "Spatial landmark detection and tissue registration with deep learning"

<https://www.nature.com/articles/s41592-024-02199-5>

Using segmentation to do CLEM registration

Zheng et al. "A Complete Electron Microscopy Volume of the Brain of Adult *Drosophila melanogaster*"

[https://www.cell.com/cell/fulltext/S0092-8674\(18\)30787-6](https://www.cell.com/cell/fulltext/S0092-8674(18)30787-6)

The authors could consider adding this citation for use of mutual information for registration:

Wells et al. "Multi-modal volume registration by maximization of mutual information"

Citation for general segmentation / landmark based multi-modal registration:

Anatomy-guided Multimodal Registration by Learning Segmentation without Ground Truth: Application to Intraoperative CBCT/MR Liver Segmentation and Registration

<https://pmc.ncbi.nlm.nih.gov/articles/PMC8184611/>

The wrong citation for BigWarp is used. From the github page, they ask to cite:

JA Bogovic, P Hanslovsky, A Wong, S Saalfeld, "Robust registration of calcium images by learned contrast synthesis", In Biomedical Imaging (ISBI), 2016 IEEE 13th International Symposium on, 1123-1126, DOI: 10.1109/ISBI.2016.7493463.

Reviewer #2 (Remarks on code availability):

I looked at the provided sample data. I installed clem-reg (went smoothly), and used it to open the sample data. I did not try to run it (because I was working on a laptop.)

It's great that the plugin allows one to open the sample data directly. It took me some time to find where the menu item was though - please consider pointing this out on your README in github.

Reviewer #3 (Remarks to the Author):

This manuscript presents CLEM-reg, a workflow for automated registration of volumetric light and electron microscopy images. The method is based on independent segmentation of both modalities, followed by probabilistic alignment of point clouds sampled from the segmented images. The performance is evaluated against manual alignment using two newly developed measures. The workflow is freely available as user-friendly napari plugin including data for testing. The source code is well organised, includes automated testing and is easy to read. Overall, the idea of automating CLEM registration by combining segmentation and point cloud registration is powerful and potentially widely applicable, and the manuscript is well written and complete. Although none of the individual steps is novel per se, they are here combined in a clever way and with an intuitive user interface. There is however room for improvement: The specific implementation, documentation and the examples given are a bit limited, but this can probably easily be addressed by the authors - more specifically:

- 1) The intended audience for the napari plugin are probably biologists and microscopists without detailed knowledge of the used algorithms and python. Novice users would benefit from more complete, step-by-step instructions on how to install the plugin, how to run the sample workflow, and how to adapt the parameters to own datasets/how to prepare own datasets. For example, in the instructions for setting up the environment, the step to activate the environment is missing. Information on how to start napari and how to open the sample data could also be added. Also, many of the parameters that can be set in the GUI are not described.
- 2) One strength of the method as claimed in the text is "the option to leverage alternative tools from other plugins or platforms". The method relies on two segmentation workflows, MitoNet for EM and a Laplacian-of-Gaussian based segmentation, which are both robust and well documented, but the workflow as presented here is thus constrained to datasets where both can be applied. In particular, the generalisability of MitoNet could be tested and discussed, given that the plugin uses the Empanada framework where in principle other trained models can be imported. The point cloud alignment step uses probreg, a well established python library for probabilistic registration. For expert users, it would be great to have a bit more specific examples (e.g. in the form of a commented Jupyter notebook) showing how the workflow could be adapted to other data to make it more generally applicable, e.g. by using other pre-trained segmentation models in Empanada or from the bioimage model zoo. In principle, the segmentation, feature detection and point cloud registration/warping are all independent from each other and could be refined/extended independently.
- 3) Line 123-124: "Since CLEM-Reg, to our knowledge, is the first algorithm to automate vCLEM alignment, manual registration

by an expert was used as a baseline for comparison." Can the authors be more specific as to why the other cited methods (e.g. ecCLEM) could not be tested on the data used here? Anyway, the data in this manuscript are freely available, which will enable such a comparison to other methods in the future. Have the authors considered running their method against some publicly available benchmark datasets, e.g. from the COMULIS project?

4) One major limitation is that the point cloud registration only converges for rigid registration, but not for the deformable registration often needed in CLEM. It would be great if there would be a bit more detail on why this probably did not work here, if it is an inherent limitation of the used algorithm or rather the datasets used. Are there other PC registration methods that could potentially work here?

5) Line 312: downsampling of the FM and EM datasets is generally required prior to execution. How large can datasets be? Maybe a performance analysis as a function of input image size would help to estimate the amount of downsampling needed.

Minor comments:

- Lines 324ff, array tomography etc., please include some references here.

- Fig. 2, a slightly more detailed depiction of how MitoNet works could be included here (or at least some annotation of the "boxes").

- The developers have answered questions on github, additionally some users have asked questions on the image.sc forum during the last few months that have not yet been answered.

Technical notes:

Installation worked both on a 2021 Macbook M1 and on a Linux GPU Workstation. On Mac, the plugin worked as described, but the EM segmentation with MitoNet was very slow (>2h). Under Linux, there was an error due to a wrong CuDNN version in the environment. After changing the CuDNN version to 8, the segmentation worked but stopped after an error about a missing "libnvfuser_codegen.so". It would be easier to troubleshoot such (quite common) problems if more details about the required dependencies and versions would be given (the requirements.txt file is empty)

Reviewer #3 (Remarks on code availability):

See main review

Version 3:

Decision Letter:

Our ref: NMETH-A56852C

19th May 2025

Dear Martin,

Thank you for submitting your revised manuscript "CLEM-Reg: An automated point cloud based registration algorithm for volume correlative light and electron microscopy" (NMETH-A56852C). It has now been seen by the original referees and their comments are below. The reviewers find that the paper has improved in revision, and therefore we'll be happy in principle to publish it in Nature Methods, pending minor revisions to satisfy the referees' final requests and to comply with our editorial and formatting guidelines.

TRANSPARENT PEER REVIEW

Nature Methods offers a transparent peer review option for new original research manuscripts submitted from 17th February 2021. We encourage increased transparency in peer review by publishing the reviewer comments, author rebuttal letters and editorial decision letters if the authors agree. Such peer review material is made available as a supplementary peer review file.

Please state in the cover letter 'I wish to participate in transparent peer review' if you want to opt in, or 'I do not wish to participate in transparent peer review' if you don't. Failure to state your preference will result in delays in accepting your manuscript for publication.

Please note: we allow redactions to authors' rebuttal and reviewer comments in the interest of confidentiality. If you are concerned about the release of confidential data, please let us know specifically what information you would like to have removed. Please note that we cannot incorporate redactions for any other reasons. Reviewer names will be published in the peer review files if the reviewer signed the comments to authors, or if reviewers explicitly agree to release their name. For more information, please refer to our <https://www.nature.com/documents/nr-transparent-peer-review.pdf> target="new">FAQ page.

ORCID

Sincerely,
Rita

Rita Strack, Ph.D.
Senior Editor
Nature Methods

Reviewer #1 (Remarks to the Author):

I would like to thank the authors for carefully addressing all my points in their revised document. In the current status, I do not have any further questions nor suggestions and I wish CLEM-Reg a fruitful life and many users in the future.
Best

Reviewer #2 (Remarks to the Author):

I found the revised manuscript to be clearer and the authors' additional analysis strengthened the conclusions and answered the questions that I and other reviewers had.

In particular, the additional comparisons of CLEM-Reg results to those of manual registration and the quantification of errors as a function of distance to mitochondria in Supp Fig 9 (that CLEM-Reg generates landmarks from) are an informative addition that gives me extra confidence in the robustness of the method.

Supp Fig 11 is great and informative visualization in my opinion; I hope the authors agree. The broad regions of very low error inside the cell boundaries are exactly what I would hope to see. Even the higher error hot spots have low error.

There is an edit to the main manuscript that had phrasing that was confusing to me, on lines 523-4:

"Note that the transformation matrix is computed in pixels, but conversion to physical units can be achieved by appropriate scaling of the translation vector."

It is not clear to me exactly what "translation vector" means in this context. The translation part of the transformation matrix?

Perhaps omitting that part of the sentence would be sufficient. Something like:

"... but conversion to physical units can be achieved by using an appropriate scaling factor."

would be clear, in my opinion.

Reviewer #2 (Remarks on code availability):

I have no additional comments on the code.

I watched the additional tutorial videos provided by the authors and found them helpful, and agree that there is a valuable addition.

Reviewer #3 (Remarks to the Author):

The authors have significantly improved and expanded the manuscript in response to the reviews. The documentation is now more complete and includes tutorial videos. The installation instructions have also been improved, and a Docker container option was added. The segmentation masks for the demo dataset are now provided to test the registration workflow. I understand that providing a generic segmentation workflow would be beyond the scope of this work, but for advanced users, it is now easier to modify the workflow accordingly to use it with different segmentation methods. The lack of a direct comparison with Autofinder is justified given the effort by the authors. The supplementary material has been amended with more details explaining the performance of the method. Overall, I see that all points have been sufficiently addressed and that publication in Nature Methods of this nice method is justified.

Reviewer #3 (Remarks on code availability):

The code is well documented and runs both on MacOS and Linux.

Version 4:

Decision Letter:

15th Jul 2025

Dear Martin,

I am pleased to inform you that your Article, "CLEM-Reg: An automated point cloud based registration algorithm for volume correlative light and electron microscopy", has now been accepted for publication in Nature Methods. The received and accepted dates will be June 27, 2024 and July 15, 2025. This note is intended to let you know what to expect from us over the next month or so, and to let you know where to address any further questions.

Over the next few weeks, your paper will be copyedited to ensure that it conforms to Nature Methods style. Once your paper is typeset, you will receive an email with a link to choose the appropriate publishing options for your paper and our Author Services team will be in touch regarding any additional information that may be required. It is extremely important that you let us know now whether you will be difficult to contact over the next month. If this is the case, we ask that you send us the contact information (email, phone and fax) of someone who will be able to check the proofs and deal with any last-minute problems.

Authors may need to take specific actions to achieve compliance with funder and institutional open access mandates.

If your research is supported by a funder that requires immediate open access (e.g. according to [Plan S principles](https://www.springernature.com/gp/open-science/plan-s-compliance) or the [NIH public access policy](https://www.springernature.com/gp/open-science/us-federal-agency-compliance)) then you should select the gold OA route, and we will direct you to the compliant route where possible. Because authors warrant under our subscription licensing terms that they haven't committed to licensing any version of their article under a licence inconsistent with the terms of our agreement – including the applicable embargo period – publication under the subscription model isn't suitable for authors whose funders require no embargo.

If you are active on Twitter/X or Bluesky, please e-mail me your and your coauthors' handles so that we may tag you when the paper is published.

Best regards,
Rita

Rita Strack, Ph.D.
Senior Editor

** Visit the Springer Nature Editorial and Publishing website at http://editorial-jobs.springernature.com?utm_source=ejP_NMeth_email&utm_medium=ejP_NMeth_email&utm_campaign=ejp_Nmeth > www.springernature.com/editorial-and-publishing-jobs for more information about our career opportunities. If you have any questions please click [here](mailto:editorial.publishing.jobs@springernature.com) . **

In response to Dr. Strack's summary comments:

In terms of the tool itself, we ask that you focus on addressing the technical concerns and questions,

1. ***especially regarding rigid vs deformable registration (ref 3) and***

This has been addressed by the addition of an extra figure (**Supp. Fig. 4**) displaying the comparison between the different transformation types

2. ***better quantifying accuracy/error (refs 1 and 2).***

This has been addressed by the addition of comparisons between the fully automated CLEM-Reg and BigWarp (manual landmarks) transforms as suggested by Reviewer #2, with results shown in **Supp. Fig. 10 and 11**.

3. ***In terms of the tool itself, we ask that you update documentation as requested by the refs and offer improved user guidance.***

Documentation and user guidance has been improved in the GitHub repository, in particular in the associated README.md file. In addition, we have provided a containerised Docker based option for installation and various tutorial videos.

4. ***Regarding the presentation, we ask that you clarify what types of EM data the method is suitable for and how the approach could be made more generally extensible.***

We clarify that the current implementation necessitates an aligned vEM stack. In the case of FIB SEM and SBF SEM, as discussed in the Materials and Methods, this is a simple routine pre-processing step to correct for a small amount of per-slice translational jitter. However, other methods, such as serial section TEM (ssTEM) and array tomography (AT), require an additional preprocessing step to correct for more extreme translational, rotational and non-linear warping and alignment, which is beyond the scope of the CLEM-Reg tool. However, we note in the discussion that the tool may provide additional leverage for (co-)optimising such methods.

5. ***We also ask that you either explore or discuss the limits of the method in terms of scale.***

We have elaborated on our discussion of current technical limitations. While point cloud based methods alleviate some bottlenecks typical for registration workflows, we note that the current implementation retains some steps where data size should be managed, for instance by using downscaled vEM data. We discuss the potential of new technologies such as next generation file formats (NGFF) and Dask in terms of allowing efficient CLEM registration across a range of scales. In addition, we estimate how the time required for registration scales with the number of input points in **Supp. Fig. 3**.

We thank the reviewers for the comments and have addressed them below.

Reviewer #1:

- **Installing the software:**

Following the proposed installation procedure is not successful. Although it looks like it should be simple, after launching Python 3.8, the command line states:

```
conda create -n clemreg_env python=3.9
File "<stdin>", line 1
conda create -n clemreg_env python=3.9
^
```

SyntaxError: invalid syntax

Please keep in mind that your end-user is a biologist with no expertise in programming. Already installing a plugin in ImageJ must be heavily documented (copy past in the proper folder), so command line is scary.

We recommend using Python 3.9 and, as outlined in our manuscript, have tested the installation of napari-clemreg on Linux and macOS. We do however agree that better installation guidance is necessary, in particular keeping in mind the end-users, and have thus added tutorial videos to the GitHub repository providing a step-by-step video guide to install the plugin (from the error message, it seems like conda might not be installed). In addition, we have provided a Docker container and associated instructions for its installation including a tutorial video for installation.

- CLEM-Reg Installation Guide: <https://youtu.be/ZN68q9OU59s>
- Docker Installation Video: <https://youtu.be/2GRB99UIP6g>

- **Transformation model:**

The registration comparison is done between an affine registration (manual landmark deposition) and a rigid registration (CLEM-Reg). For an accurate comparison, I would have expected the same transformation model to be proposed. BigWarp (Bogovic et al., 2016) proposes a rigid transformation through its 'similarity' transformation type. Alternatively, an affine transformation form could be proposed in CLEM-Reg. If this is already proposed, then I am sorry if I missed that part.

The napari-clemreg plugin does in fact have the options for rigid, affine and non-linear (thin-plate spline) transformations. However, we observed better registration accuracy when using rigid alignment (**Supp. Fig. 4**) and thus chose the transformation model that worked best in each case. We have clarified the transformation options in the manuscript in the section "Generating modality-agnostic point clouds from internal landmarks and registration".

- The spline line model is indeed often requested by the community as it seems intuitively more accurate but happens to be counterproductive in many cases. This was already carefully studied by (Paul-Gilloteaux et al., 2017) and (Bogovic et al., 2016) and a direct reference to their work could support this coherent result.

We thank the reviewer for pointing us to the two references which we have included in the manuscript. We added the sentence “While non-rigid registration is often thought to deliver superior alignment performance, both the results obtained here with CLEM-Reg and prior work by others (14,40) do not support this assumption.” to the discussion.

- **Model orientation:**

I understand the benefits of resampling the dataset into a similar orientation for a human expert to achieve good registration: our brain is meant to identify and extrapolate morphological similarities and cannot easily navigate 3D volume with orthogonal orientations. However, is this also necessary for a computer, working on a cloud of points with statistical robustness? Although it seems irrelevant to the material used in this article, it would be a game changer in volume registration of tissue imaged with even more modalities (Karreman et al., 2014). If the authors could discuss (demonstrate?) this point, it would be interesting.

The crucial preprocessing step here is to mirror one of the volumes, since there is no fixed convention on the “handedness” of coordinate systems across different imaging modalities. Without this step, it is impossible for some transformation types to converge. For all of our benchmark vCLEM dataset, we found it necessary to perform this step.

While the reviewer is in principle correct that under ideal conditions we would expect there to be no difference regardless of the initial alignment. However, in practice, the point clouds are noisy and can thus end up converging to a local minimum, i.e. an incorrect solution that minimises the error sufficiently for the algorithm to finish running. For instance, Zhang *et al.* [1] show that the CPD algorithm works best when starting from within a 50-60° initial orientation. For practical purposes, this can be achieved by pre-aligning to the nearest orthogonal orientation, which is a relatively computationally cheap operation and requires no pixel interpolation. In situations where an approximate relative alignment is known a priori, such as CLEM, applying this simple step is more efficient than a brute force search of multiple starting configurations.

- **Density of points:**

The results demonstrate the benefits of down sampling the number of points on the outer membrane of the segmented structures on the calculation time, without impacting the registration accuracy. Does this relate to an absolute number of points? Or a density per volume? That could be the beginning of a solid benchmarking tool for the future algorithms that the community will build upon CLEM-Reg. eC-CLEM illustrated the limited benefit of adding points beyond a certain number, by monitoring the 'Target Registration Error'. In practice, beyond 15 points in 2D or 25 points in 3D, if they can even be identified, the registration precision is sufficient to accurately localize a structure below the diffraction limit. I would appreciate if the authors could elaborate on that aspect (minimal number, if it even exists).

The points placed in eC-CLEM are inherently different from the points in the point clouds sampled by CLEM-Reg. In the first case, these points are manually placed in precise locations and correspondence is known between pairs of points across modalities. In the latter case, points are a memory-efficient representation of the segmentation masks automatically obtained from both modalities. Therefore, the absolute number of points would not necessarily be a useful metric, perhaps a more important heuristic is the distribution of these points. For this purpose, we recommend choosing a widespread structure to provide the best alignment, demonstrated with mitochondria (spread widely through single cell images). In tissue, for example, nuclei (distributed throughout the volume) may be an appropriate choice.

- **Imaging resolution:**

One key benefit of manual registration is the sub-pixel point placement, that relies on the subjective expert experience to identify the centroid of the homology structure, regardless of the resolution in the three axes for each dataset. I feel that this relates to the discrepancy in accuracy that decays faster with pixel size binning than with points removal, regardless of the computing time.

Along this line, what would be the impact of a lower resolution by light microscopy, especially in Z, on the final registration accuracy? The dataset presented here is using state-of-the-art light microscopy apparatus, but we can expect also light microscopy datasets of lower resolution (epifluorescence, laser scanning confocal, cryo-LM). Could that cause registration failures? it's an open question for discussion.

To quantitatively address this point, we explored if using unprocessed FM data acquired on an Airyscan (i.e. without applying Airyscan post-processing and thus equivalent to a confocal stack in terms of its optical resolution) would perform similarly to using post-processed Airyscan FM data as used throughout the paper. Side-by-side comparisons of unprocessed confocal-like and processed Airyscan images are shown in **Supp. Fig. 6**. We perform quantification on segmented lysosomes and endosomes shown in **Supp. Fig. 7 and 13** (EM volume overlaid by FM signal and centroid distances between segmented lysosomes/endosomes). We find that the overlays do not meaningfully differ when using confocal-like or Airyscan FM data and that CLEM-Reg therefore also performs with near-expert accuracy on confocal-like FM data.

It is of note that the initial developments of CLEM-Reg and in particular the FM segmentation part, were performed on epifluorescence microscopy data (mitochondrial channel only). With regards to other modalities, the prerequisite for successful registration is the ability to segment common landmarks across modalities, the downstream processing of the point clouds is agnostic to the origin of the points.

- **A fundamental mathematic question that does not need to be addressed in the manuscript:**

Although I doubt my question is mathematically meaningful (unpaired cloud of points), do you think you could extrapolate a registration error prediction, similar to what was done on the ‘leave-one-out’ (Kukulski et al., 2011) or ‘Target Registration Error’ (Paul-Gilloteaux et al., 2017)? The use of cloud-point might prevent that as a statistically relevant result. A direct response is sufficient and not need to be discussed in the paper as it might blur the message. Unless the author thinks it is supporting the general purpose of the manuscript.

This is an interesting point. However, we are hesitant to include a measure of “accuracy”, since it could easily be misinterpreted. In the case of the two referenced papers, it should be noted that this approach works because the points are linked and it is therefore known *a priori* which points map to each other. Based on this known correspondence, registration errors can be computed by leaving out pairs of points systematically and computing the resulting error. This is however not the case for our approach, since points are sampled from structures without knowing how they are related. Thus, we cannot use the mentioned approach to estimate errors.

Minor comments:

- Line 83-85: The manual registration is not responsible for introducing bias: the use of the adequate structure for registration is the responsibility of the scientist. Should I have multiple channels to conduct the registration, I would follow your strategy of registering independent labels, even manually. Please rephrase.

This statement has been softened in the Introduction to “[...] since the target structures may be directly used for registration”.

- 156-159: from the text, it seems that you clean up the segmented mask by a simple size base filter. However, in the figure 2A, 3D segmentation (top right panel) looks slightly different from Z=7 (bottom left panel). As if you added a ‘dilate’ step on the mask. Is it only due to a differential Z selection in the figure? Or is there a missing step in the image processing description?

We are impressed by the reviewer’s acute attention to detail! Indeed, the segmentation mask on the top of **Fig. 2A** was different from the one showing different Z slices. The top row was intended as a graphical illustration for the overall processing and thus we did not ensure that exactly the same mask was used to show the insets. We have now corrected this using the same segmentation mask consistently throughout the figure. Note that the top row shows a 3D rendering of the segmentation mask throughout the volume.

- Line 173: fixed voxel size of 10? What unit? I assume nanometre. The unit is also missing on supp figure 1.

This has been clarified in the section “Generating modality-agnostic point clouds from internal landmarks and registration.” to “For instance, reducing the point sampling frequency from 1/16 to 1/256 with a fixed voxel size of 10x10x10 pixels (points within each voxel are averaged to generate exactly one point) leads to [...]”.

We note that voxel size in terms of physical units depends on the pixel-size of the image volumes provided by the user.

- Line 190: the rigid registration outperforms the non-linear alignment... ‘data not shown’? I don’t need to have it in the paper as it would impair readability, but please mention it’s not shown if that’s the case.

We have included additional quantification supporting this statement in **Supp. Fig. 4** and referenced it where appropriate.

- Line 196: I would not state that it occupies both the centre and the periphery. One could argue about the centre of the dataset or the sample. Across dispersed areas of the cells seems accurate enough.

We have rephrased it as the reviewer suggested. The sentence in the section “Assessing CLEM-Reg performance against experts” has been changed to “The selected lysosomes varied in size (0.029 μm^3 to 0.522 μm^3) and were distributed across the cell volume [...]”.

- Line 222-223: where are located the more accurately registered lysosomes? Are they at the periphery? at the centre? What could account for that? The transformation model (rigid/affine) or the distance to the centre of the dataset? Or is it just random? This should be discussed.

We computed the Spearman correlation between centroid distances of off-target structures (lysosomes or endosomes) with the distance to mitochondria. The distance to mitochondria was obtained by computing a distance transform on the mitochondria segmentations at the centroid of each off-target structure. Only in the first dataset (EMPIAR-10819) could we observe a slight correlation between alignment performance of lysosomes/endosomes and distance to mitochondria ($\rho=0.5$). This has been added as a new supplementary figure, **Supp. Fig. 9**. An additional paragraph was added to the section “Assessing CLEM-Reg performance against experts” discussing the results.

- Line 337: ‘those structures are numerous and distributed...’ I already commented on the number of points required (absolute number or density threshold), but I am also interested in the distribution parameters. eC-CLEM elaborated significantly on this aspect, illustrating the importance of homogeneous distribution. I assume this is comparable. A 2D projection of the manual landmark would be appreciated.

We have added a projection of the manual landmarks, with cell outlines for context to the supplement as **Supp. Fig. 11**.

- Line 459: voxel-size of $s=15$ what? Unit?

We have clarified in the text that this refers to units of pixels. The sentence was adapted to: “A voxel-size of $s=15 \times 15 \times 15$ pixels was used for all datasets.”

Comment on the figures:

- Figure 1C: could the channels be converted to black and white for better visibility? The green channel is barely visible in the received document. I would also suggest increasing the dpi of the figure, so the lines of the GUI become readable. On a printed version, it is not readable at all. Furthermore, in the user manual of the software, I would strongly recommend giving a detailed description for each parameter independently: an ‘interpolation order’ for ‘image warping’ does not mean much to most biologist.

We have improved the resolution and the colours in **Fig. 1A and C** to improve readability/visibility. We have also updated the README of the GitHub repository to provide a more detailed explanation of each of the parameters.

- Supplementary figure 1 (**now Supp. Fig. 2**): A- on target channel (title) or off-target channel (legend)? Same question for B. A and B: Voxel size 10, 20, 40, 80 (what unit?).

We have updated the legend for consistency and added units to clarify.

- Supplementary note line 19-20: I’m not sure to understand the binning part. If many points are within the same binning area, they become 1, regardless of the number? Meaning that anything ranging from 0 to X points is turning into 0 or 1 whitening that voxel? Could you elaborate more precisely in the text?

The positions of the points within each voxel are implicitly taken into account, since point positions are averaged to generate exactly one point within each voxel. An additional explanation of how points are binned was added to the section “Generating modality-agnostic point clouds from internal landmarks and registration”: “[...] with a fixed voxel size of 10x10x10 pixels (points within each voxel are averaged to generate exactly one point) leads to [...]”. The corresponding supplementary note was also updated with the sentence: “[...] while the second parameter bins the point cloud by a given voxel size (pixels) such that points within each voxel are averaged to generate exactly one point.”

- Supplementary figure 3 (**now Supp. Fig. 12**): same comment as for figure 1C. LM in black and white, and larger screenshots of the split workflow would help readability (at least in the user manual).

We thank the reviewer for this suggestion and have now added larger images for each of the steps of the split registration workflow to help readability.

- Supplementary figure 7A (now **Supp. Fig. 1A**): Robustness to miss-labelling is an important result when it comes to 40% loss! I understand that the figure gets in supp data, but I would recommend empathizing it in the discussion. Personal comment: this is not the scope of the paper, but this might be used by MitoNet to improve training of their software (especially on SBF datasets) in a feedback loop.

We have included a sentence in the results section “Segmenting internal landmarks” to better highlight this result: “The robustness of CLEM-Reg to missing mitochondria segmentations in the EM volume was estimated by randomly removing segmentations (Supp. Fig. 1A). The registration accuracy of CLEM-Reg was constant up to a loss of around 40% of segmented mitochondria in EM.”

We also included additional comments in the discussion to highlight this point. The sentence in the Discussion on limitations due to segmentation errors has been adapted to take this comment into account. It now reads: “Nevertheless, certain limitations remain in regard to the automated organelle segmentation in vEM which adversely impacts the alignment performance when more than 40% of mitochondria are missed during the segmentation step (Supp. Fig. 1A).”

We also thank the reviewer for the comment on MitoNet fine-tuning - indeed we speculate that iterative optimisations may give leverage to improve both the segmentation and registration processes and have added this to the discussion.

- Supplementary figure 7B (now **Supp. Fig. 1B**): observing that loss of peripheral landmarks is impacting more significantly the registration accuracy than central elements is coherent with already published work, see previous comment (Paul-Gilloteaux et al., 2017).

We thank the reviewer for highlighting this reference and have cited it accordingly. A sentence was added to the discussion to draw further attention to this result: “Registration performance with CLEM-Reg is also more sensitive to the loss of peripheral landmarks (Supp. Fig. 1B) which is consistent with prior work by Paul-Gilloteaux *et al.* (40).”

Supporting materials

- Could the author provide video tutorials, similar to what is available on BigWarp or eC-CLEM? This would be certainly helpful to the end-user and support fast

spread of the tool to the community. Installing the software and third-party elements; running the software natively, importing external landmarks etc.

We have added video tutorials for the installation and usage of CLEM-Reg to the GitHub repository:

- CLEM-Reg Installation Guide: <https://youtu.be/ZN68q9OU59s>
- Docker Installation Guide: <https://youtu.be/2GRB99UIP6g>
- CLEM-Reg Tutorial: <https://youtu.be/ud3zTLgl8Ks>
- CLEM-Reg Split Registration Tutorial: <https://youtu.be/cypDti0UUwY>

Reviewer #2:

Evaluation

- Since an expert has already clicked landmark pairs for these data as part of this evaluation, the authors could use the transformation estimated by clem-reg to transform the moving landmark placed by the expert and measure its distance to the corresponding point placed manually. This would give a measure of agreement with the manual method for every placed landmark.

We thank the reviewer for this insightful comment, and have performed the suggested analysis. Additional quantification comparing manual registration by an expert to CLEM-Reg on manually placed landmarks was performed and results shown in Supp. Fig. **10A-C**. Additional analysis exploring the robustness of these results to random errors in manual landmark placement were carried out in Supp. Fig. **10D**. A paragraph was added to the section “Assessing CLEM-Reg performance against experts” discussing these results.

- Even further, it is possible to sample points in your moving image and 1) transform them with the clem-reg transformation 2) transform them with the manual transformation 3) compute their distance, to create a dense "heat map" of errors over space. This may or may not be that informative, but it should be straightforward to try.

We thank the reviewer for this insightful comment, and have performed the suggested analysis. Dense heat-maps were obtained by computing the Euclidean distance between landmarks placed in EM and landmarks placed in LM transformed by CLEM-Reg. Results are shown in **Supp. Fig. 11**.

Related questions:

- How many landmarks were placed by the expert using BigWarp? What kind of transformation was used in BigWarp? (Figure 4's caption suggests affine - was that 12 degrees of freedom?). Please clarify this. Related - does the transformation for the "rigid" CPD have 9 degrees-of-freedom? If there is a difference, then it could affect how to interpret this evaluation.

Details on the number of landmarks placed was added to the section “Assessing CLEM-Reg performance against experts”: “Next, performance of CLEM-Reg against manual registration was quantified on manually placed landmarks in the

LM and EM volumes using BigWarp (EMPIAR-10819: number of point pairs n=145, EMPIAR-11537: n=42 and EMPIAR-11666: n=55).”

The affine transform computed with BigWarp had 12 degrees-of-freedom while the rigid transform computed with CPD had 7 degrees-of-freedom (rotation, translation and one scaling factor). It was found that rigid CPD outperformed affine CPD on the datasets used in this study. However, to ensure that performance quantifications reflected how BigWarp is typically used by end-users, the affine transform from BigWarp was chosen.

- Using mitos as a fiducial makes sense, especially given the success of mito-net. I wonder about the accuracy of registration far from mitochondria though. Do the authors know if results are similarly good throughout the volume - especially far from mitos? It would be interesting to know if this has been tested. Or, if there is uncertainty and the authors agree that there is a risk that results are worse far from mitos, then it would be good to make researchers working on such structures aware of this. From supplementary Figure 2, it appears that the evaluated lysosomes are all close to mitochondria, but it is difficult to know.

We thank the reviewer for the comment and have addressed this question. Distances of target landmarks (endosomes and lysosomes) to mitochondria segmentations were correlated to centroid distances computed for the registration performance quantification in **Supp. Fig. 9**. An additional paragraph was added to the section “Assessing CLEM-Reg performance against experts” discussing the results.

- Supp Fig 7B (now **Supp. Fig. 1B**) may be informative when it comes to this question, given the "central / intermediate / peripheral" labels. Qualitatively, it appears that removing intermediate and peripheral mitos has a larger effect. I would be interested to hear the authors' interpretation of this, if they have ideas.

Removing intermediate and peripheral has larger effects which, as pointed out by Reviewer #1, was also found by Paul-Gilloteaux *et al.* [2]. Intuitively, errors in rotational alignment cause absolute errors in landmark placement proportional to the distance from the rotation centre. Since the rotational centre for these errors is likely centralised amongst the points, it follows that such errors dominate at larger distances, in the intermediate and peripheral regions.

Minor Questions

- line 169:
"extraction of coordinate of pixels"
Are the transformations computed in pixel or physical units?

We have clarified in the text that transformations are computed in pixels. A sentence was added to the methods section ("Point cloud downsampling and registration with CPD and BCPD") to clarify this point: "Note that the transformation matrix is computed in pixels, but conversion to physical units can be achieved by appropriate scaling of the translation vector."

- line 189:
do they authors have an idea as to why rigid outperformed non-linear alignment?
This would be an interesting

This is addressed in a paragraph in the discussion and is consistent with prior work by Paul-Gilloteaux *et al.* [2] and Bogovic *et al.* [3]: "[...] While non-rigid registration is often thought to deliver superior alignment performance, both the results obtained here with CLEM-Reg and prior work by others (14,40) do not support this assumption.

A possible explanation for this counterintuitive finding is that better performance can be achieved by restricting the degrees of freedom, since the point-cloud data from the two modalities are inherently noisy and do not perfectly match. Thus, as noise increases due to factors like segmentation errors, non-isotropic pixel-sizes in the FM volume or point sampling, the non-linear registration approach is more prone to converging to non-optimal solutions due to the larger parameter space. Rigid registration restricts the degrees of freedom and thus the parameter space which facilitates convergence towards an optimal solution."

- line 199:
does the reported time include the time needed for segmentation of mitos?

Yes, the segmentation time is included. We have added clarification to the manuscript.

Citations

- Using Deep learning and landmarks:

Ekvall et al. "Spatial landmark detection and tissue registration with deep learning"

<https://www.nature.com/articles/s41592-024-02199-5>

- Using segmentation to do CLEM registration
Zheng et al. "A Complete Electron Microscopy Volume of the Brain of Adult *Drosophila melanogaster*"
[https://www.cell.com/cell/fulltext/S0092-8674\(18\)30787-6](https://www.cell.com/cell/fulltext/S0092-8674(18)30787-6)
- The authors could consider adding this citation for use of mutual information for registration:
Wells et al. "Multi-modal volume registration by maximization of mutual information"
- Citation for general segmentation / landmark based multi-modal registration:
Anatomy-guided Multimodal Registration by Learning Segmentation without Ground Truth: Application to Intraoperative CBCT/MR Liver Segmentation and Registration
<https://pmc.ncbi.nlm.nih.gov/articles/PMC8184611/>
- The wrong citation for BigWarp is used. From the github page, they ask to cite:
JA Bogovic, P Hanslovsky, A Wong, S Saalfeld, "Robust registration of calcium images by learned contrast synthesis", In
Biomedical Imaging (ISBI), 2016 IEEE 13th International Symposium on, 1123-1126, DOI: 10.1109/ISBI.2016.7493463.

We thank the reviewer for these suggestions and have referenced the paper by Wells *et al.* in the introduction. We have also corrected the citation for BigWarp.

Remarks on code availability:

- It's great that the plugin allows one to open the sample data directly. It took me some time to find where the menu item was though - please consider pointing this out on your README in github.

We have added more detailed instructions to the README. In addition, we have added a compressed TIFF of a MitoNet segmentation to the repo, as "notebooks/data/em_mask.tif" for situations where a prospective user might want to test the workflow without immediate access to a GPU for the MitoNet segmentation step.

Reviewer #3:

- The intended audience for the napari plugin are probably biologists and microscopists without detailed knowledge of the used algorithms and python. Novice users would benefit from more complete, step-by-step instructions on how to install the plugin, how to run the sample workflow, and how to adapt the parameters to own datasets/how to prepare own datasets. For example, in the instructions for setting up the environment, the step to activate the environment is missing. Information on how to start napari and how to open the sample data could also be added. Also, many of the parameters that can be set in the GUI are not described.

We thank the reviewer for this suggestion which was also raised by the other reviewers. We have recorded video tutorials and extended the GitHub README to provide end-users with in-depth explanations of how to use the plugin.

- CLEM-Reg Installation Guide: <https://youtu.be/ZN68q9OU59s>
 - Docker Installation Guide: <https://youtu.be/2GRB99UIP6g>
 - CLEM-Reg Tutorial: <https://youtu.be/ud3zTLgl8Ks>
 - CLEM-Reg Split Registration Tutorial: <https://youtu.be/cypDti0UUwY>
- One strength of the method as claimed in the text is "the option to leverage alternative tools from other plugins or platforms". The method relies on two segmentation workflows, MitoNet for EM and a Laplacian-of-Gaussian based segmentation, which are both robust and well documented, but the workflow as presented here is thus constrained to datasets where both can be applied. In particular, the generalisability of MitoNet could be tested and discussed, given that the plugin uses the Empanada framework where in principle other trained models can be imported. The point cloud alignment step uses probreg, a well established python library for probabilistic registration. For expert users, it would be great to have a bit more specific examples (e.g. in the form of a commented Jupyter notebook) showing how the workflow could be adapted to other data to make it more generally applicable, e.g. by using other pre-trained segmentation models in Empanada or from the bioimage model zoo. In principle, the segmentation, feature detection and point cloud registration/warping are all independent from each other and could be refined/extended independently.

We thank the reviewer for this excellent suggestion. We have now added example Jupyter notebooks highlighting how CLEM-Reg can be run headless and using other algorithms in each step. In addition, we have included a

segmentation mask for the EM segmentation (notebooks/data/em_mask.tif in the GitHub repository) to allow testing without a GPU.

- Line 123-124: "Since CLEM-Reg, to our knowledge, is the first algorithm to automate vCLEM alignment, manual registration by an expert was used as a baseline for comparison." Can the authors be more specific as to why the other cited methods (e.g. ecCLEM) could not be tested on the data used here? Anyway, the data in this manuscript are freely available, which will enable such a comparison to other methods in the future. Have the authors considered running their method against some publicly available benchmark datasets, e.g. from the COMULIS project?

We tested ec-CLEM Autofinder as suggested by the reviewer. On our data, the point finder identified suitable points on the mitochondrial channel of the FM, but did not find suitable points on the EM, as expected. This is why we included EM segmentation using MitoNet in our pipeline. However, we were unable to run the Autofinder plugin after point detection due to a "NoSuchMethod" error. We also tested Autofinder with the EM segmentations produced by CLEM-Reg/Mitonet, but encountered another "NoSuchMethod" error. After talking to the developer, we were advised to use "ec-CLEM v2", which is in beta-testing and has not been released yet. We tried to install it following the instructions on GitHub, but failed at step 2, since the instructions were insufficient for non Java specialists/biologists.

However, it is important to note that Autofinder has been developed to register Quantum dots, detected by the point finder in the FM and manually annotated in the EM. Quantum dots are not widely used for CLEM. We prefer not to use them in our cells as it may disrupt their metabolism. Disregarding the errors we encountered, the requirement to provide EM segmentations or manual landmarks to run Autofinder makes it a bottleneck for our CLEM data, which we addressed by including MitoNet in our pipeline.

Regarding the use of other publicly available vCLEM data like COMULIS, we want to point out that the datasets used in the "COMULISglobe 3D-CLEM" challenge (see screenshots) are in fact our own benchmark datasets (shown in **Fig. 4B and 6B**). We are not aware of other openly available vCLEM datasets.

TASK 4: COMULISglobe 3D-CLEM

Title: Cellular level Volume Electron Microscopy (EM) – Light microscopy (LM)

Illustration:

First row: 2d slices of one of the released datasets (raw EM data, raw LM data, EM rigidly registered on LM data).
Second row: 3D views of the same volume.

Training/Validation/Test: Download

Description:

Automatic multimodal microscopy 3D image registration is an unsolved problem in image processing. The aim of this first challenge, organized by the COMULISglobe society, is to set up the basis of a recurrent challenge. Electronic microscopy EM 3D image data – Focused Ion Beam Scanning Electron (FIB SEM) and Serial Block Face Scanning Electron Microscopy (SBF SEM) – were captured on the same cell area as light microscopy LM 3D image data (Super resolution fluorescence microscopy). They were acquired with variable volume sizes (~15000 × ~15000 × ~10000 for isotropic raw EM data and ~2000 × ~2000 × ~100 for anisotropic raw LM data) and field of views (approx. 75 × 75 × 50 micrometers³). The in-plane resolution was constant (isotropic voxel size of 0.005 micrometers for EM, 0.035 micrometers in xy and 0.13 micrometers for light microscopy).

EM data is non specific and shows all organelles. LM data is specific and is composed of two color channels: the first one showing mitochondria, the second one showing the nuclei of the cells.

Number of cases: The total number of datasets is three. Two data sets are used for training/validation, and the third dataset for testing. For this challenge the datasets were cropped to patches: 40 training/validation, 20 testing.

Annotation: Manually annotated landmarks in all datasets will be used for validation and testing.

Pre-processing: Common pre-processing to same voxel resolutions and spatial dimensions as well as rigid pre-registration will be provided to ease the use of learning-based algorithms for participants with little prior experience in image registration.

License: EM Data are released under the CC0 license, and LM data and landmarks under the CC-BY-NC 4.0 license.

Citation:

[1] Daniel Krentzel, Matouš Elphick, Marie-Charlotte Domart, Christopher J. Peddie, Romain F. Laine, Ricardo Henriques, Lucy M. Collinson, Martin L. Jones. CLEM-Reg: An automated point cloud based registration algorithm for correlative light and volume electron microscopy. bioRxiv 2023.05.11.540445; doi: <https://doi.org/10.1101/2023.05.11.540445>

<https://www.comulis.eu/>

Screenshots from <https://learn2reg.grand-challenge.org/>

- One major limitation is that the point cloud registration only converges for rigid registration, but not for the deformable registration often needed in CLEM. It would be great if there would be a bit more detail on why this probably did not work here, if it is an inherent limitation of the used algorithm or rather the datasets used. Are there other PC registration methods that could potentially work here?

The non-rigid algorithm does converge, but the resulting overlay is not as good as the one obtained from rigid CPD when applying the performance metrics developed in this paper. This initially came as a surprise to us, but it is consistent with findings from prior work (e.g. Paul-Gillouteaux *et al.* [2]). Additional quantification supporting this statement has been added in **Supp. Fig. 4** and referenced where appropriate.

- Line 312: downsampling of the FM and EM datasets is generally required prior to execution. How large can datasets be? Maybe a performance analysis as a function of input image size would help to estimate the amount of downsampling needed.

We thank the reviewer for this comment. In principle, data set size corresponds to the number of points used during registration. Once we are in the point cloud space, the size of the input images becomes irrelevant, since the coordinates can take arbitrarily large values as long as the number of points is the same (**Supp. Fig. 3** shows the relationship between the number of points and the time

required for registration). This means that scaling the coordinate values by e.g. a factor of 100 does not change the memory requirements. However, obtaining the segmentations from which points are sampled will require more compute which in turn necessitates more working memory, unless individual chunks are processed sequentially and stored on disk (one way of addressing this could be by enabling users to work with a next generation file format like OME-Zarr once a stable version is released). This reduces memory requirements, but the time will scale by the number of chunks that need to be processed. Image warping after registration will also require more compute as dataset size increases (the napari-clemreg plugin supports chunked image warping which was used to generate the overlays for the non-linear BCPD). The time required to warp the moving volume thus also scales linearly with the number of chunks that are warped. The time required to segment and warp images as a function of image size therefore scales cubically, while the time required for registration as a function of the number of points scales by a factor of 1.47 to 1.69 (**Supp. Fig. 3**).

Minor comments:

- Lines 324ff, array tomography etc., please include some references here.

We thank the reviewer for the comment and have added references.

- Fig. 2, a slightly more detailed depiction of how MitoNet works could be included here (or at least some annotation of the "boxes").

Details of the MitoNet architecture have been added by annotating the architecture and the figure caption has been updated accordingly.

- The developers have answered questions on github, additionally some users have asked questions on the image.sc forum during the last few months that have not yet be answered.

The question raised on image.sc was addressed via private interactions with the person in question. We will make sure to resolve this on the image.sc forum.

Technical notes:

- Installation worked both on a 2021 Macbook M1 and on a Linux GPU Workstation. On Mac, the plugin worked as described, but the EM segmentation with MitoNet was very slow (>2h). Under Linux, there was an error due to a wrong CuDNN version in the environment. After changing the CuDNN version to

8, the segmentation worked but stopped after an error about a missing "libnfvfuser_codegen.so". It would be easier to troubleshoot such (quite common) problems if more details about the required dependencies and versions would be given (the requirements.txt file is empty)

MitoNet running very slowly on the M1 mac is likely due to driver compatibility issues. To ease the usage of the plugin across platforms we have provided a Docker image with an accompanying tutorial video detailing how to use it. The empty requirements.txt is a left-over file from prior versions of the source code and has been removed. The required packages can be found in the "setup.cfg" file.

- Docker image: <https://github.com/krentzd/napari-clemreg/blob/main/clemreg.dockerfile>
- Docker Installation Guide Video: <https://youtu.be/2GRB99UIP6g>
- Setup file: <https://github.com/krentzd/napari-clemreg/blob/main/setup.cfg>

References:

- [1] Zhang, P., Qiao, Y., Wang, S., Yang, J. and Zhu, Y., 2017. A robust coherent point drift approach based on rotation invariant shape context. *Neurocomputing*, 219, pp.455-473.
- [2] Paul-Gilloteaux, P., Heiligenstein, X., Belle, M., Domart, M.C., Larijani, B., Collinson, L., Raposo, G. and Salamero, J., 2017. eC-CLEM: flexible multidimensional registration software for correlative microscopies. *Nature methods*, 14(2), pp.102-103.
- [3] Bogovic, J.A., Hanslovsky, P., Wong, A. and Saalfeld, S., 2016, April. Robust registration of calcium images by learned contrast synthesis. In *2016 IEEE 13th international symposium on biomedical imaging (ISBI)* (pp. 1123-1126). IEEE.